# A disassembly-driven mechanism explains F-actin-mediated chromosome transport in starfish oocytes

**Philippe Bun[†], Serge Dmitrieff[‡], Julio M Belmonte, François J Nédélec\*, Péter Lénárt\***

Cell Biology and Biophysics Unit, European Molecular Biology Laboratory, Heidelberg, Germany

**Abstract** While contraction of sarcomeric actomyosin assemblies is well understood, this is not the case for disordered networks of actin filaments (F-actin) driving diverse essential processes in animal cells. For example, at the onset of meiosis in starfish oocytes a contractile F-actin network forms in the nuclear region transporting embedded chromosomes to the assembling microtubule spindle. Here, we addressed the mechanism driving contraction of this 3D disordered F-actin network by comparing quantitative observations to computational models. We analyzed 3D chromosome trajectories and imaged filament dynamics to monitor network behavior under various physical and chemical perturbations. We found no evidence of myosin activity driving network contractility. Instead, our observations are well explained by models based on a disassembly-driven contractile mechanism. We reconstitute this disassembly-based contractile system *in silico* revealing a simple architecture that robustly drives chromosome transport to prevent aneuploidy in the large oocyte, a prerequisite for normal embryonic development.

DOI: https://doi.org/10.7554/eLife.31469.001

**\*For correspondence:**
nedelec@embl.de (FJN);
lenart@embl.de (PL)

**Present address:** [†]IPNP, INSERM U894, Paris, France; [‡]CPN, INSERM U894, Paris, France

**Competing interests:** The authors declare that no competing interests exist.

## Introduction

Muscle contraction forms the basis of animal locomotion, and at the microscopic scale, contraction of actin filament (F-actin) networks drives a plethora of essential cellular processes. Most prominently, the cortex, a thin and entangled network of F-actin underlying the plasma membrane determines the shape of animal cells, and changes in cell shape underlie cell migration, cell division and tissue morphogenesis (*Munjal and Lecuit, 2014*; *Pollard and Cooper, 2009*; *Salbreux et al., 2012*). While myosin motor activity, non-muscle myosin II in particular, is the main driver of these processes, filament dynamics has been shown to play a critical role as well. For example, cell migration is to a large part driven by polymerization of actin filaments at the cell front and depolymerization at the rear (*Cramer, 2013*; *Mseka and Cramer, 2011*; *Pollard and Borisy, 2003*; *Ridley, 2011*). Cytokinetic ring closure has been shown to be driven by depolymerization of filaments in some species, although myosin activity certainly is the main driver in others (*Green et al., 2012*; *Mendes Pinto et al., 2012*; *Neujahr et al., 1997*).

In addition to its cortical functions, recent studies revealed extensive 3D F-actin networks in the bulk cytoplasm of large oocytes and embryos with essential functions in intracellular transport processes (*Field and Lénárt, 2011*). For example, mouse oocytes feature a dynamic cytoplasmic F-actin network required to position the nucleus (*Almonacid et al., 2015*), the meiotic spindle (*Almonacid et al., 2014*; *Azoury et al., 2008*; *Schuh and Ellenberg, 2008*), and to transport vesicles from the cell center to the cortex, including cortical granules to prevent polyspermy (*Cheeseman et al., 2016*; *Schuh, 2011*). While myosin II and myosin V motors have been involved in transporting the spindle and vesicles, respectively, filament dynamics appear to have a key role in

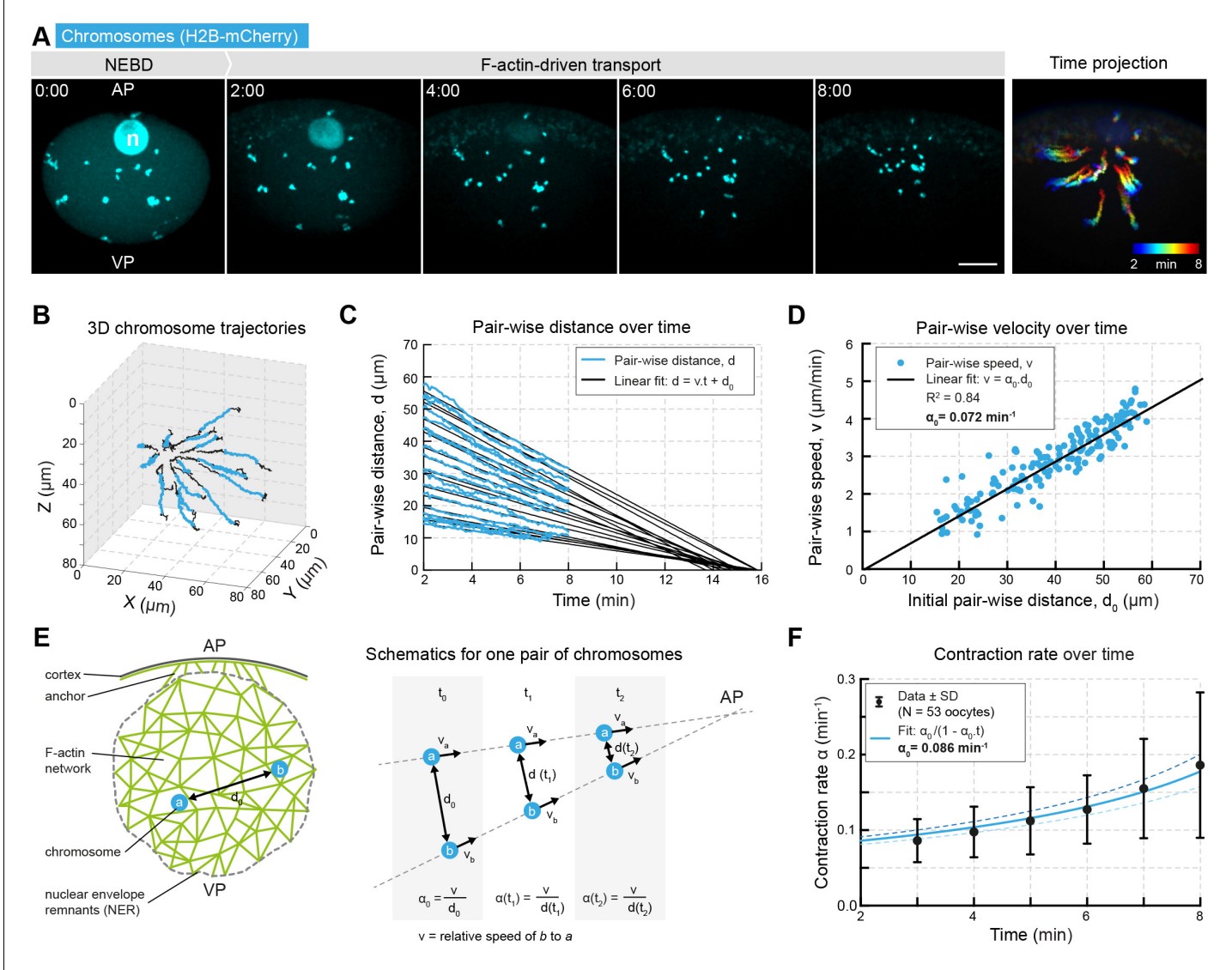

**Figure 1.** Network contraction is characterized by a single rate. (**A**) Maximum z-projections of selected time points through the nuclear region of live starfish oocytes expressing H2B-mCherry (cyan) to label chromosomes during actin-driven chromosome congression. n: disassembling nucleolus; AP: animal pole; VP: vegetal pole; right: pseudo-color time projection of z-projections. (**B**) 3D plot of chromosome trajectories derived from the data set in (**A**) during chromosome congression (2–8 min after NEBD highlighted in cyan). (**C**) Plot of pair-wise distances of chromosomes, d versus time for the same data set. The pair-wise approach speed was determined by a linear fit, as shown, and extrapolated to visualize 'congression time'. (**D**) Pair-wise chromosome approach speeds depend linearly on the initial distance, $d_0$ for all chromosome pairs. (**E**) Schematics for one pair of chromosomes while being transported to the AP with a constant speed and decreasing distance. Thus, the contraction rate, $\alpha$ is increasing through the process. (**F**) Contraction rates calculated for 2 min intervals from 2 to 8 min after NEBD and fitted with *Equation 3* to determine the initial approach rate, $\alpha_0$. Best fit $\alpha_0$ is shown in solid line, $\alpha_0 \pm 5\%$ is shown in dashed lines. Scale bars, 20 µm; time is given as mm:ss relative to NEBD. N indicates the number of oocytes.

DOI: https://doi.org/10.7554/eLife.31469.002

The following figure supplement is available for figure 1:

**Figure supplement 1.** Contraction is independent of anchoring at the animal pole.
DOI: https://doi.org/10.7554/eLife.31469.003

these processes as well, regulated by Fmn2/Spire1/2 as well as cortical Arp2/3 nucleators, and other factors (*Chaigne et al., 2013*; *Cheeseman et al., 2016*; *Holubcová et al., 2013*; *Pfender et al., 2011*; *Schuh, 2011*; *Yu et al., 2014*).

Generally, while the mechanism of contractile force generation is well understood in sarcomeric-like assemblies, such as muscles or stress fibers, the mechanisms in disordered F-actin networks, such as the above quasi-2D cortical and 3D cytoplasmic networks, remains much more elusive (*Lenz et al., 2012*; *Murrell et al., 2015*). Additionally, computational models and theoretical work so far focused largely on disordered F-actin networks formed by non-dynamic filaments inspired by in vitro reconstituted contractile systems composed of stable filaments (*Alvarado and Koenderink, 2015*; *Belmonte et al., 2017*; *Bendix et al., 2008*; *Köhler and Bausch, 2012*; *Murrell and Gardel, 2014*). While it is true that these non-dynamic systems contract efficiently, it is also clear that filament dynamics has key contributions in vivo, as illustrated above. Indeed, although much less investigated, theoretical work has shown that F-actin dynamics is able to generate not only protrusive but also contractile stress (*Sun et al., 2010*; *Zumdieck et al., 2007*).

In the past, we discovered a prominent example of F-actin-driven intracellular transport in starfish oocytes, whereby a 3D F-actin network collects chromosomes scattered in the large oocyte nucleus and transports them to the forming microtubule spindle (*Lénárt et al., 2005*). This F-actin network forms in the nuclear region after nuclear envelope breakdown (NEBD), at the onset of meiosis, and is essential to prevent formation of aneuploid eggs (*Mori et al., 2011*). It contracts homogeneously and isotropically mediating long-range and size-selective transport of chromosomes sterically trapped in the network. While the contraction of the network is isotropic and homogeneous, connections of the network to the cell cortex provide directionality, resulting in transport towards the microtubule spindle, located at the cortex (*Mori et al., 2011*).

However, the mechanism underlying contraction of this 3D F-actin network remained unknown. To reveal it, we monitored contractile behavior using the chromosomes as natural probes combined with quantitative imaging of filament dynamics in live oocytes. In parallel, we developed physical models of different mechanisms of contraction, and compared the predictions of these models to the experimental system challenged by physical and chemical perturbations. We find that observations are well fitted by models in which contraction is driven by F-actin disassembly, but not by models of motor-driven contractility. Comparisons of experimental observations to in silico reconstruction of this contractile system reveal a novel and remarkably simple architecture to transport chromosomes, to prevent aneuploidy in the large oocyte.

## Results

### Network contraction is characterized by a single scalar contraction rate

As we showed previously, chromosomes are sterically entrapped and thereby transported by a contracting F-actin network (*Mori et al., 2011*). Chromosomes can thus be used as probes to monitor the network's contractile behavior without potential artifacts associated with F-actin labeling. Therefore, we optimized the spatial and temporal resolution to image and track H2B-3mEGFP or -mCherry labeled chromosomes in 3D during actin-driven transport, 2 to 8 min after NEBD (*Monnier et al., 2012*; *Mori et al., 2011*) (*Figure 1A,B*). We reported previously that contractile activity is isotropic and while the network normally contracts to the animal pole, this merely results from passive anchoring of the network to the cell cortex (*Mori et al., 2011*). Thus, here we analyzed the pair-wise distances between chromosomes to characterize the network contraction rate, a measure that is independent of the location and even existence of the anchor (*Figure 1—figure supplement 1*).

The trajectories revealed that any pair of chromosomes, irrespective of their initial location, exhibited a constant approach velocity towards one another during transport (*Figure 1C*). Extrapolation of these linear trajectories showed that all chromosomes would meet at nearly the same time and location, if the network were to contract all the way to a single point (*Figure 1C*). This implies that pair-wise chromosome velocities depend linearly on their initial separation distance (*Figure 1D*), indicative of a spatially homogeneous and isotropic contraction (*Mori et al., 2011*). This means that the ratio ($\alpha_0$) between pair-wise velocities ($v$) and *initial* pair-vise distance ($d_0$) is constant for all pairs of chromosomes:

$$\alpha_0 = \frac{v}{d_0} \qquad (1)$$

In theory, $\alpha_0$ could be simply derived as the slope of the fitted line on *Figure 1D*. In practice, however, we found the initial pair-wise chromosome separation distance difficult to determine, as

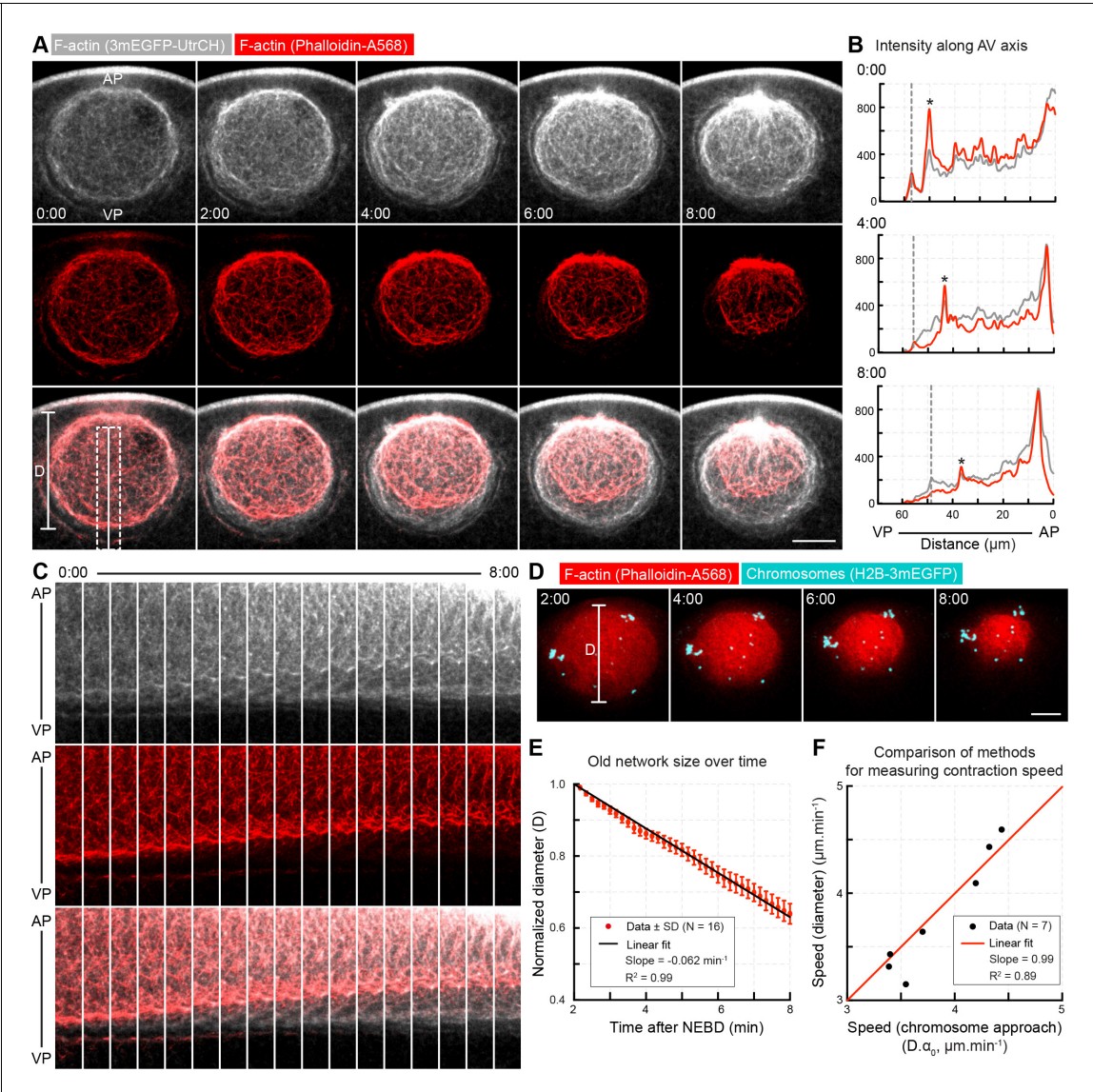

**Figure 2.** Concomitant with network contraction new filaments polymerize along its boundary. (**A**) A pulse of phalloidin-AlexaFluor 568 (red) was injected into the nuclear region ~2 min after NEBD in an oocyte expressing 3mEGFP-UtrCH (grays) to label the population of F-actin present at the time of injection. AP: animal pole; VP: vegetal pole. Selected sum-intensity z projections are shown. (**B**) Intensity profiles along the dashed line in (**A**) (2 µm wide, rolling average of 0.5 µm). Gray dashed lines indicate the position of the nuclear boundary; asterisks mark the edge of the phalloidin-labeled old network. (**C**) Kymograph-like plots of the region marked by a dashed rectangle in (**A**). (**D**) Similar to (**A**) except that chromosomes were additionally labeled by H2B-3mEGFP (cyan) in oocytes injected with phalloidin-AlexaFluor 568 (red). z-projections of selected time points are shown. (**E**) The diameter of the pulse labeled network (D as shown on (**D**)) decreases linearly during the F-actin-driven chromosome transport. Data were collected from five independent experiments. (**F**) Contraction speeds calculated by measuring network diameter corresponding to contraction speeds calculated from chromosome approach. Data were collected from two independent experiments. N indicates the number of oocytes. Scale bar: 20 µm; time is given as mm:ss relative to NEBD.

DOI: https://doi.org/10.7554/eLife.31469.004

motions caused by the collapse of the nuclear envelope at NEBD occur at the onset of network contraction. Therefore, we instead determined $\alpha_0$ by extrapolating its value (*Figure 1E,F*). Generally, at any time $t$ for any one pair of chromosomes the contraction rate $\alpha(t)$ is given by:

$$\alpha(t) = \frac{v}{d(t)} \tag{2}$$

as the speed is constant, $d(t) = d_0 - vt$, leading to:

$$\alpha(t) = \frac{\alpha_0}{1 - \alpha_0 t} \tag{3}$$

Thus, we calculated the pair-wise approach rate, $\alpha(t)$ on 2 min intervals throughout F-actin-driven transport. As expected, it increases over time according to *Equation 3*, allowing us to determine the value of $\alpha_0$ (*Figure 1F*). The value of $\alpha_0$ derived by this method from a large number of oocytes provided a precise estimate of the remarkably invariable initial contraction rate; $\alpha_0 = 0.085 \pm 0.017$ min$^{-1}$ (N = 53 ± S.D.). Note that the reciprocal of this value, $t_0 = 1/\alpha_0 = 11.76 \pm 0.27$ min is the 'convergence time', the time that would be required to transport the chromosomes from their initial position to the final meeting point.

Taken together, analysis of chromosome trajectories as probes embedded in the contracting network reveals a remarkably regular behavior indicative of a homogeneous and isotropic contraction characterized by a single parameter, the initial contraction rate, $\alpha_0$.

## As the network contracts, new filaments polymerize at its boundary

To visualize F-actin dynamics underlying this very regular contractile behavior, we expressed the fluorescent F-actin marker UtrCH (mEGFP3-UtrCH or mCherry3-UtrCH) in oocytes (*Burkel et al., 2007*). In addition, at the time of NEBD we spiked in a small amount of fluorescent phalloidin in a different color. Phalloidin bound rapidly and irreversibly to filaments present at the time of injection allowing us to distinguish the initial network present at NEBD from F-actin that polymerize later (*Figure 2A*).

Analysis of these pulse-label recordings revealed that while the network in the nuclear region contracts, new filaments are polymerized on nuclear envelope remnants (NER) forming a new network filling up the space around the contracting initial network (*Figure 2A,C*, *Video 1*). Intensity profiles showed that polymerization occurred predominantly at the boundary defined by NER, and to a limited extent within the network (*Figure 2B*). Correspondingly, we distinguish the network initially filling the nuclear space at NEBD, referred hereafter as 'old network', from the 'new network' that is polymerized from the NER boundary after NEBD.

To quantify the rate of contraction of the old network, we fitted a circle to the phalloidin-labeled region and recorded its diameter, D over time (*Figure 2E*). This revealed that the old network contracted to approximately half of its initial size, and with a nearly constant speed. The rate of contraction derived this way was similar, although somewhat lower than the rate $\alpha_0$ determined by tracking chromosomes (compare *Figures 1F* and *2E*). We then performed direct comparisons on a single cell basis combining phalloidin pulse-labeling with labeling chromosomes (*Figure 2D*). We found the contraction rate extracted from the pair-wise chromosome approach and the rate derived by mea-

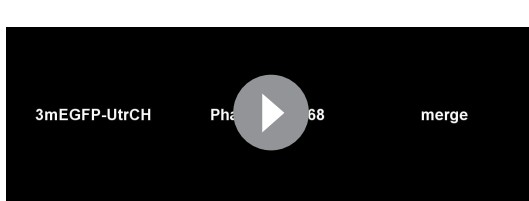

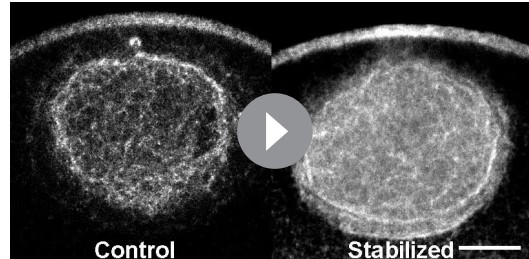

**Video 1.** Pulse labeling of the contracting F-actin network. Oocyte expressing 3mEGFP-UtrCH (gray) was injected with phalloidin-AlexaFluor 568 to label the population of F-actin present at the time of injection. Scale bar: 20 μm.
DOI: https://doi.org/10.7554/eLife.31469.005

**Video 2.** The response of the F-actin network to 3D laser ablation. 3D laser ablation was performed in oocytes expressing mCherry3-UtrCH without (left) or with recombinant UtrCH-AlexaFluor 568 nm (right) injection. Scale bar: 20 μm.
DOI: https://doi.org/10.7554/eLife.31469.008

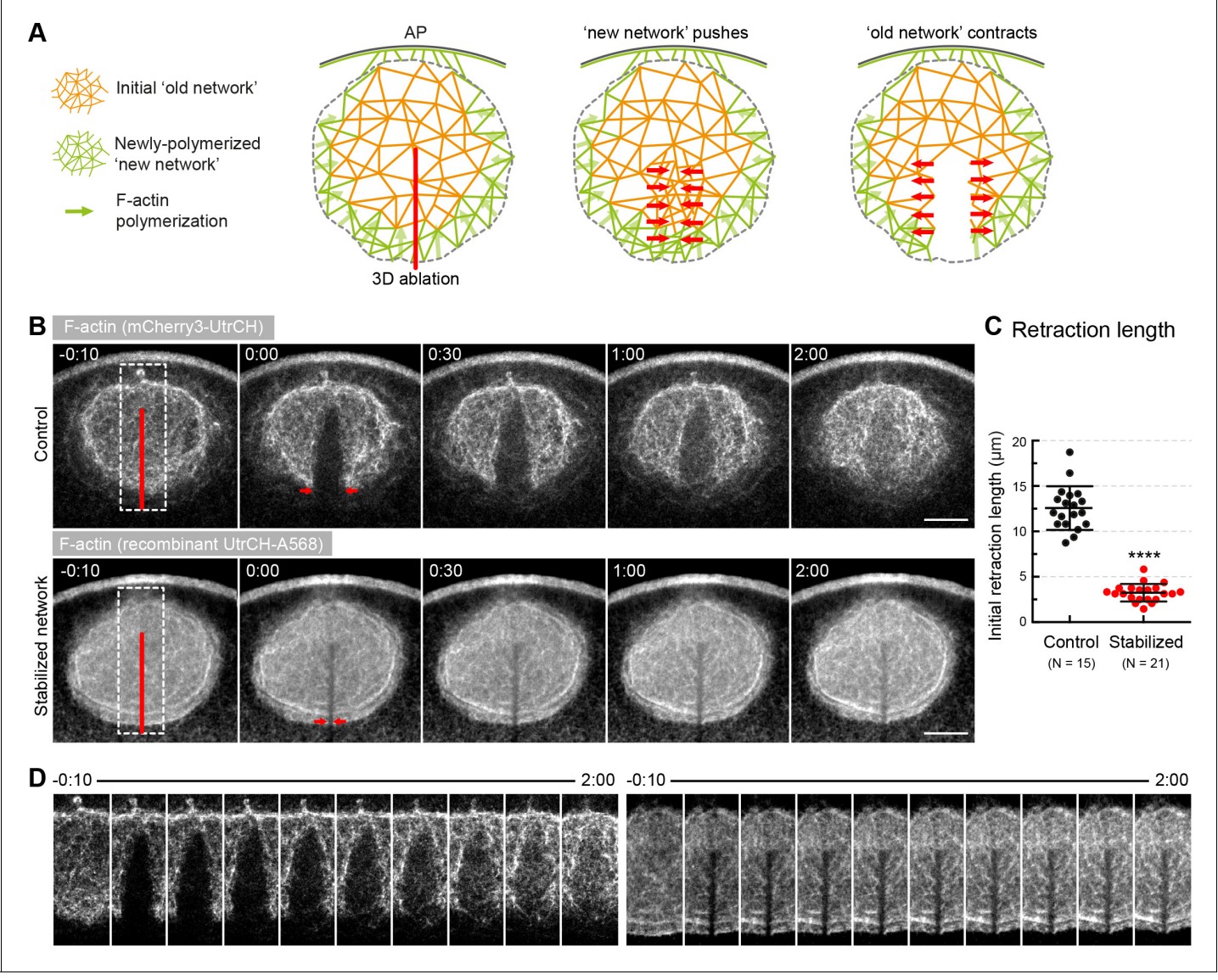

**Figure 3.** Forces in the network, rather than pushing by polymerization generate contractility. (**A**) Schematics illustrating the expected response to ablation in case of a 'pushing' mechanism vs. active contraction of the network. (**B**) Selected sum-intensity z-projections through the nuclear region of live oocytes expressing 3mCherry-UtrCH. Top row: untreated control; bottom row: injected with a large amount of recombinant UtrCH to stabilize F-actin. One frame just before and frames after 3D ablation along the animal-vegetal axis (red line) are shown. (**C**) Initial retraction distance as marked by red arrows in (**B**) in control and stabilized oocytes. Data were collected from three independent experiments. N indicates the number of oocytes. Mann-Whitney's test, ****p<0.0001. (**D**) Cut regions outlined by dashed boxes in (**B**) are shown zoomed and for several time points. Scale bar, 20 μm. Time is given as mm:ss relative to NEBD.
DOI: https://doi.org/10.7554/eLife.31469.006

The following figure supplement is available for figure 3:

**Figure supplement 1.** Effects of laser ablation are independent of the direction of the cut.
DOI: https://doi.org/10.7554/eLife.31469.007

suring the diameter of the phalloidin-labeled 'old network' to correlate precisely (*Figure 2F*). Thus, similar contraction rates are derived by the two methods, however, it appears that phalloidin used to visualize the old network slightly slows contraction and increases variability, likely dependent on the amount of phalloidin injected.

Altogether, visualizing filament dynamics identified two major components in this contractile system: the contraction of the initial 'old network' and the polymerization of the 'new network' along

the NER boundary, filling the space as the old network contracts. Chromosomes are embedded and transported by the old network, thus the contraction rate, $\alpha_0$ derived from pair-wise chromosome approach is a direct measure of the old network's contraction rate.

## Active forces within the network, not pushing by polymerization drives contraction

Having identified the two main activities, polymerization at the NER and contraction of the network, we next wondered whether chromosome transport is driven by polymerization compressing the network, or contractile forces generated within the network. To distinguish these two possibilities, we ablated the F-actin network in a 3D volume using a pulsed infrared laser. We expected the network to collapse into the ablated region, were pushing forces generated by polymerization to dominate. Instead, a recoil response would evidence tension and thus active contractile forces within the network (*Figure 3A*).

We observed a prominent recoil response with the network rapidly retracting away from the ablated region (*Figure 3B–D*, *Video 2*). This shows that the network is under tension and suggests that pushing by polymerization does not have a major contribution. After retraction, the gap did not close, but was filled progressively by F-actin suggesting that in response to the perturbation, filament assembly may occur throughout the network (*Figure 3B,D*). The recoil response did not depend on the direction of the cut, consistent with isotropic contraction (*Figure 3—figure supplement 1*).

To test that the recoil observed after laser ablation is not an artifact, we also performed this experiment in oocytes where F-actin was stabilized by injecting a large amount of recombinant, fluorescently labeled UtrCH. Similar to mouse oocytes (*Holubcová et al., 2013*), high concentration of UtrCH effectively stabilized the network and prevented its contraction (*Figure 3B–D*, *Video 2*). Stabilization also prevented the recoil response confirming the specificity of laser ablation.

We thus conclude that production of F-actin at the NER boundary is not the major driver of contraction. Instead, our data show that active forces generated within the network drive contractility. Importantly, the dramatic effects of UtrCH stabilization, along with the minor slowing effect of phalloidin injection above, suggest that F-actin dynamics has a key role in contractile force generation.

## Motor- and disassembly-driven models can both explain contraction, but predict differential response to perturbation

To investigate a possible role for filament dynamics in contraction, we quantified changes in filament amounts in space and time. We used the fluorescence intensity of mEGFP3-UtrCH, which binds to filamentous actin with high affinity, as a proxy for filament mass (*Figure 4A*) (*Belin et al., 2014*; *Burkel et al., 2007*; *Chaigne et al., 2013*; *Singh et al., 2017*). To distinguish the 'old' and 'new' networks we assumed the old network to be a sphere with a radius that decreases according to the contraction rate derived from chromosome approach (*Figure 4A*):

$$R(t) = R_0[1 - \alpha_0\, t].  \tag{4}$$

Such quantifications extrapolated to the 3D volume revealed a continuous decrease of F-actin mass within the old network through the contraction process (*Figure 4B*, for F-actin densities see *Figure 4—figure supplement 1A*). To relate this measure to the rate of filament disassembly, we expressed the mass of F-actin as the number of filaments per unit volume, and defined the rate of disassembly, $k(r,t)$ as:

$$k = k_0 \frac{1 + e^\lambda}{1 + e^{\lambda/l}}  \tag{5}$$

where $k_0$ is the initial rate of F-actin disassembly, $\lambda$ the dispersion of the length distribution of F-actin and $l(r,t)$ the normalized average length of F-actin (*Figure 4B,C*, see *Supplementary file 1* for details). We found that *Equation 5* fitted well the evolution of F-actin mass allowing us to extract the values of $k_0$ and $\lambda$ (*Figure 4B,C*). For untreated oocytes, we found $k_0$ = 0.157 and $\lambda$ = 0.6. These parameters are dimensionless as neither the length nor the number of filaments are known in absolute terms.

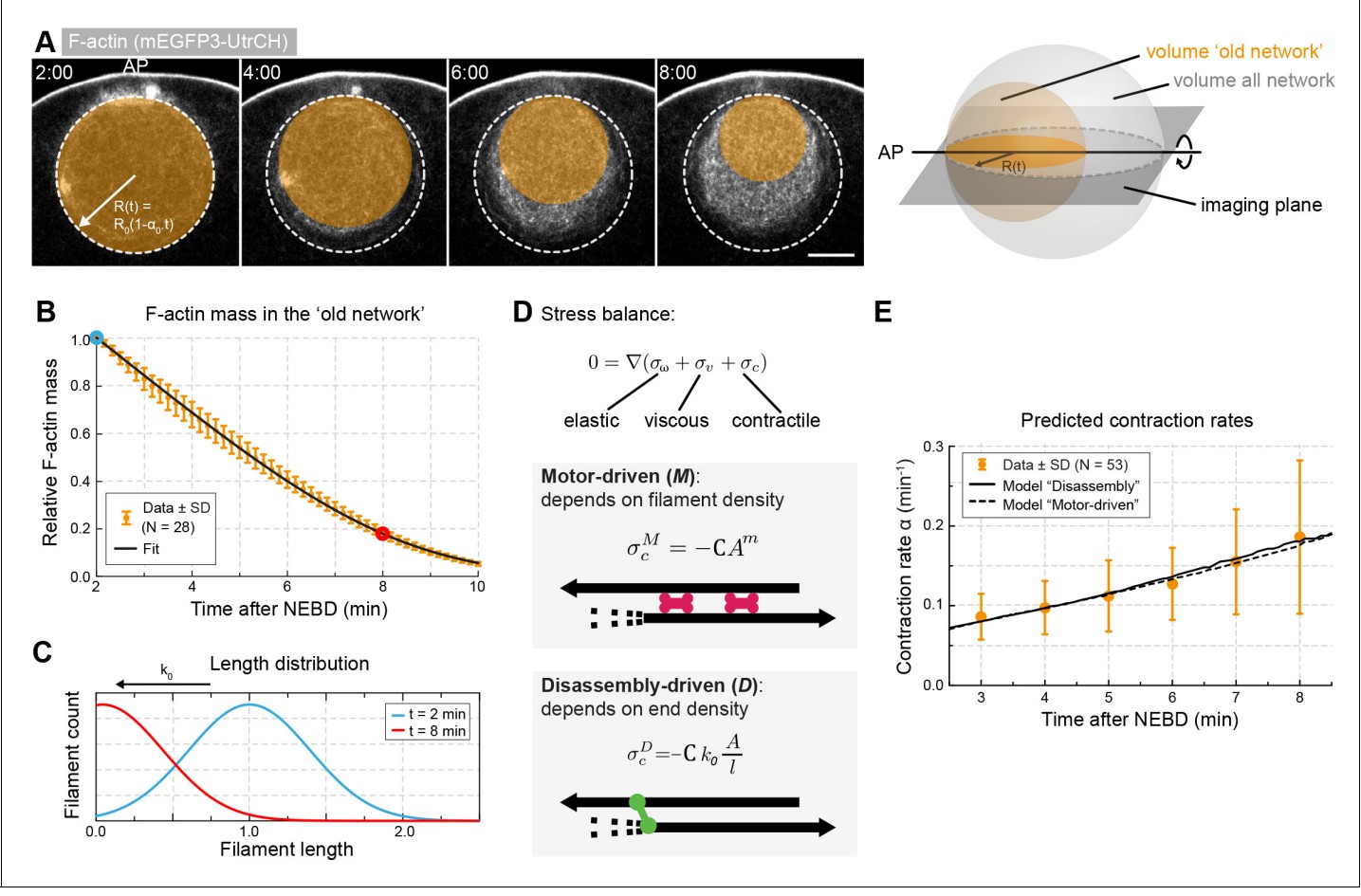

**Figure 4.** Network contraction is correlated with its disassembly. (**A**) Quantification of F-actin mass by measuring 3mEGFP- or 3mCherry-UtrCH intensities in the region corresponding to the old network (orange circle). Right: scheme illustrating extrapolation of the intensities measured in the imaging plane to the 3D volume. (**B**) Relative changes in F-actin mass (normalized between 0 and 1) over time calculated as explained in (**A**) for several oocytes. *Equation 5* was used to fit the data (black curve). Data were collected from five independent experiments. (**C**) Filament size distribution at the start (cyan) and end (red) of the contraction process: filament length decreases as set by the depolymerization rate $k_0$. (**D**) Schematic representations and equations for the two viscoelastic models, motor-driven (**M**) and disassembly-driven (**D**). (**E**) Fits of the two models, D in continuous line, M: dashed line to the contraction rates measured experimentally as shown in *Figure 1F*. N indicates the number of oocytes. Scale bar, 20 μm; time is given as mm: ss relative to NEBD.

DOI: https://doi.org/10.7554/eLife.31469.009

The following figure supplement is available for figure 4:

**Figure supplement 1.** F-actin densities in the old network.

DOI: https://doi.org/10.7554/eLife.31469.010

Next, we explored potential mechanisms that can explain contraction under the constraints of the observed filament dynamics. F-actin networks are commonly modeled as active viscoelastic gels (*Hannezo et al., 2015*; *Hawkins et al., 2009*; *Julicher et al., 2007*; *Kruse et al., 2006*; *Marchetti et al., 2013*). For example, this approach was successfully used to model F-actin contraction in droplets of *Xenopus* egg extracts, with size and time scales comparable to our experimental system (*Lewis et al., 2014*). Thus, we developed a similar model assuming that effects of filament orientation can be neglected (i.e. assuming isotropic stresses), which is justified especially when no large-scale ordering of the system is observed (*Hannezo et al., 2015*; *Hawkins et al., 2014*; *Lewis et al., 2014*). We further assumed that viscosity mainly arises from network rearrangements rather than viscous drag, or in other words, that the main factor opposing contraction is the unbinding kinetics of cross-linkers (*Blanchoin et al., 2014*; *Salbreux et al., 2012*).

We modeled the F-actin network as a viscoelastic gel in which viscous stress $\sigma_v$ (originating from friction within the network) and elastic stress $\sigma_\omega$ (originating from F-actin deformation) oppose the network contractile stress $\sigma_c$. We further generalized the model to allow viscosity and elasticity of the system to depend on F-actin concentration, noted A.

We assumed the *elastic stress* to be isotropic and neo-Hookean (*Lewis et al., 2014*):

$$\sigma_\omega = G_0\, A^g (1 - \omega) \tag{6}$$

in which $\omega$ is the strain and $G_0\, A^g$ is the elastic modulus, with $g = \{1\,,\,2\}$ characterizing the power law dependence of elasticity on A.

The *viscous stress* is written as:

$$\sigma_\nu = \bar{\eta}_0\, A^\mu \left( \nabla \otimes v + \nabla \otimes v^T \right) \tag{7}$$

where $\bar{\eta}_0\, A^\mu$ is the viscosity, with $\mu = \{1\,,\,2\}$ being the power law dependence of viscosity on A.

For an F-actin gel, the *contractile stress* in a *motor-driven process* is generally written as (*Hannezo et al., 2015*; *Julicher et al., 2007*; *Marchetti et al., 2013*; *Prost et al., 2015*):

$$\sigma_c^M = -C\, A^m \tag{8}$$

where $C$ is the contraction coefficient and $m = \{1\,,\,2\}$ is the power law dependence of contraction on A (*Figure 4D*).

Unlike in a motor-driven process, we expect F-actin *disassembly* to generate a contractile stress dependent on the density of filament ends, and on their disassembly rate. Given that $A/l$ corresponds to the density of ends and $k$ the disassembly rate, the contractile stress can be written as (*Figure 4D*; see *Supplementary file 1* for details):

$$\sigma_c^D = -C\, k_0\, \frac{A}{l} \tag{9}$$

At these scales, we assume inertia to be negligible, thus the gradient of stresses is zero:

$$\nabla(\sigma_\nu + \sigma_\omega + \sigma_c) = 0 \tag{10}$$

We determine the velocity field $v(r)$ of network displacement, which is set by force balance, by integrating *Equation 10* in space. Then, we obtain $A(r,t)$, $l(r,t)$ and $\omega(r,t)$ by integrating the equations of mass balance over time:

$$\frac{\partial A}{\partial t} + \nabla.(vA) = -k\frac{A}{l} \tag{11}$$

$$\frac{\partial l}{\partial t} + \nabla.(vl) = -k \tag{12}$$

$$\frac{\partial \omega}{\partial t} + \nabla.(v\omega) = \gamma_\omega \frac{k}{l}(1 - \omega) \tag{13}$$

in which $\gamma_\omega\, k/l$ is the strain relaxation rate. We integrated *Equations 10, 11, 12 and 13* numerically (see *Supplementary file 1* for details), and from the solution $v(r,t)$, calculated $\alpha(t)$ using *Equation 2*. Since we could not find an analytical solution to the system of equations, we are unable to provide a simple formula for $\alpha(t)$ that is derived from first-principles. In addition, we defined the parameter $\epsilon$ describing adhesion at the nuclear boundary, such that the adhesive stress at the boundary is $-\epsilon\, C\, A^m$ (see *Supplementary file 1*).

In the above framework of the active-gel theory, we first investigated whether a motor-driven (referred hereafter as model M) or a disassembly-mediated process (referred hereafter as model D) could possibly account for the observed decrease in F-actin mass during contraction. We determined the viscoelastic parameters by randomly sampling parameter values and fitting the integration results to the contraction rate $\alpha(t)$ measured experimentally (*Table 1* and *Figure 1F*). We find that both models can recapitulate the observed behavior (*Figure 4E*). However, the models additionally

**Table 1.** Dimensionless viscoelastic parameters for untreated oocytes.

| Parameter name | Symbol | Model M | Model D |
|---|---|---|---|
| Elasticity | $G_0/k_0\bar{\eta}_0$ | 2.1264 | 1.4778 |
| Contractility | $C/k_0\bar{\eta}_0$ | 1.1961 | 1.3082 |
| Adhesion to boundary | $\epsilon$ | 0.6232 | 0.7838 |
| Strain relaxation factor | $\gamma_\omega$ | 9.5898 | 2.3754 |
| Elastic power law exponent | $g$ | 2 | 1 |
| Contractile power law exponent | m | 2 | 1 |
| Viscous power law exponent | $\mu$ | 1 | 1 |

DOI: https://doi.org/10.7554/eLife.31469.011

predict the contraction rate to depend directly on disassembly in model D, whereas this dependence is indirect and additionally depends on motor activity for model M. Thus, it is expected that the dependence of contraction rate on disassembly rate will differentiate the two models. Motivated by this prediction, we experimentally manipulated motor activity and the disassembly rate $k_0$ to assess whether model M or D can predict the observed contraction rate $\alpha_0$ in various perturbation experiments.

## Perturbations of non-muscle myosin II have no effect on the rate of contraction

If the contractile process is motor-driven, non-muscle myosin II (NMYII) is the prime candidate to mediate it, as myosin II is the only myosin type that is able to assemble into minifilaments and generate force by pulling filaments towards one another (*Murrell et al., 2015*; *Sellers, 2000*; *Syamaladevi et al., 2012*). Indeed, in whole-oocyte proteomics analyses we found NMYII to be abundant in the starfish oocyte (data not shown). Thus, to test the possible role of NMYII we inhibited its motor activity by the small-molecule inhibitor, blebbistatin. Further, we over-expressed myosin regulatory light chain (MRLC), which we found in starfish oocytes to enhance NMYII activity (*Bischof et al., 2017*). To our surprise, these perturbations did not affect contraction rates derived from pair-wise chromosome approach (*Figure 5A,B*). To confirm the effectiveness of the treatments, we followed the same oocytes until the end of the first meiotic division, when the polar body is extruded, accompanied by a global surface contraction wave driven by NMY-2 (*Bischof et al., 2017*). Consistent with our earlier results, the amplitude of the contraction wave was significantly decreased upon blebbistatin treatment, while it was increased in oocytes over-expressing MRLC (*Figure 5C,D*). Thus, chromosome transport and derived contraction rates were unaffected by NMY-2 perturbations, which were effective as assessed by their effects on surface contraction waves in the same oocytes. Consistently, we saw no effects on chromosome transport following inhibition of the two common pathways of NMYII activation through myosin light chain kinase and Rho kinase, by injecting ML-7 and Y-27632, respectively (*Figure 5A–D*).

The only alternative scenario we could envisage is a mechanism similar to that driving long-range vesicle transport, nuclear and spindle positioning in mouse oocytes (*Chaigne et al., 2013*; *Holubcová et al., 2013*; *Schuh, 2011*). In this case, directional transport is thought to result from polymerization of actin filaments on vesicles by Fmn2/Spire1/2 nucleators and transport of vesicles on these filaments by myosin V motors. However, a similar mechanism is unlikely to function is starfish oocytes, as this mechanism is strictly dependent on vesicles and in the nuclear region of starfish oocytes we did not observe any membranous structures (*Figure 5—figure supplement 1A*), and the time scales do not match with the process running an order of magnitude slower in mouse as compared to starfish oocytes. Additionally, when we expressed a dominant-negative tail construct of myosin Vb (*Schuh, 2011*), we observed no effect on chromosome transport and derived contraction rates (*Figure 5—figure supplement 1B,C*).

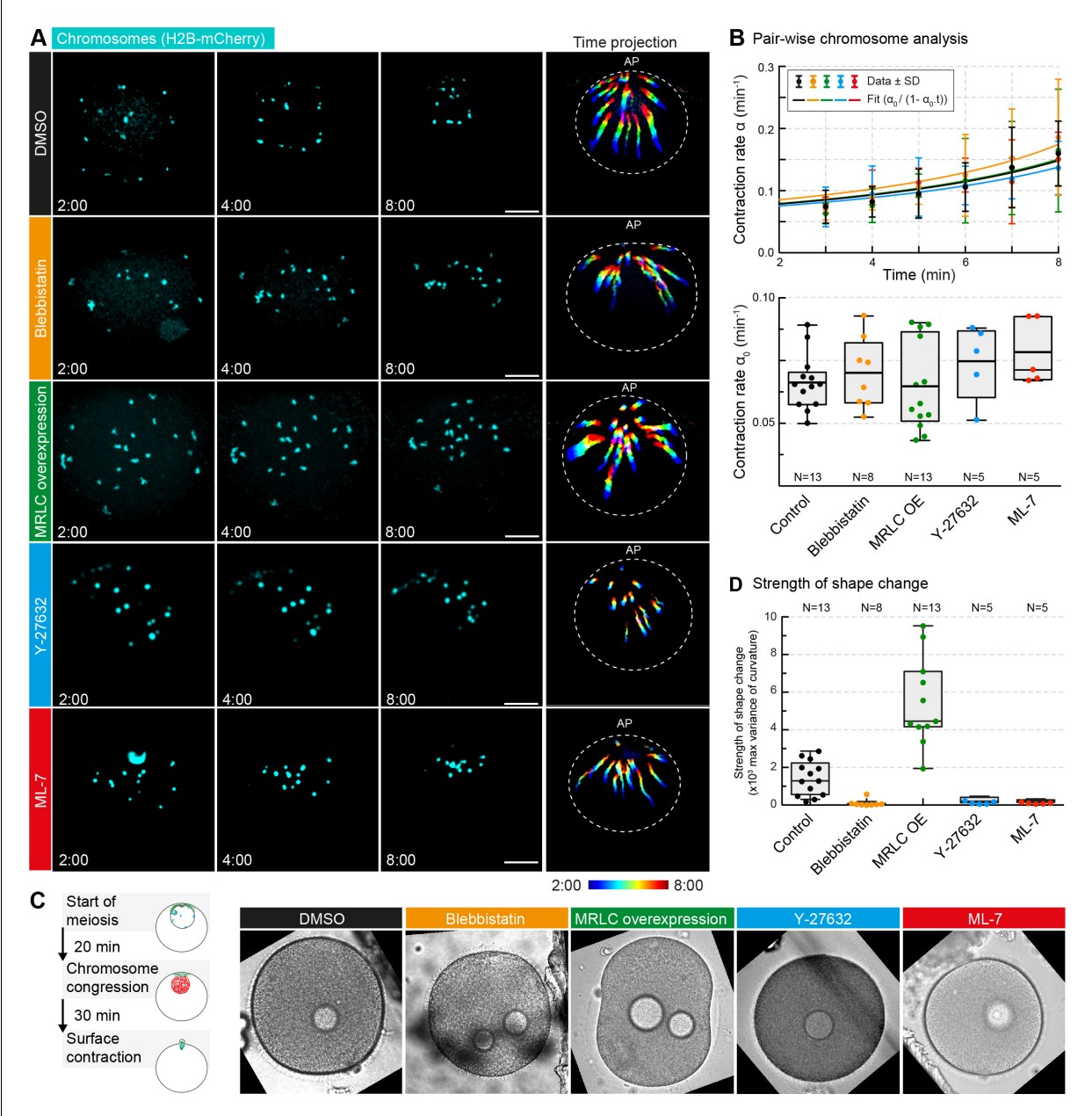

**Figure 5.** Non-muscle myosin II perturbations do not affect the rate of contraction. (**A**) Maximum-intensity z-projection through the nuclear region of oocytes expressing H2B-mCherry (cyan), either incubated for 3 hr with blebbistatin (300 µM), for 1 hr with ML-7 (100 µM) or Y-27632 (100 µM), or expressing MRLC-mEGFP, or treated with DMSO at a concentration corresponding to that of the blebbistatin treatment. Dashed circles represent the initial nuclear contour. Right: pseudocolored time projection of z-projections. Scale bar, 20 µm; time is given as mm:ss relative to NEBD. (**B**) Top: contraction rate over time for each condition with fits for determining $\alpha_0$ done as for **Figure 1F**. Bottom: box plots combined with dot plots of derived values of $\alpha_0$ for multiple oocytes. N indicates the number of oocytes. ANOVA: $p < 0.26$. (**C**) Left: schematic of starfish meiosis to illustrate the relative times of NEBD, chromosome congression and surface contraction waves. Right: transmitted light frames at the maximum deformation during the surface contraction wave of oocytes treated exactly as for (**A**). (**D**) Quantification of the strength of the shape change (maximum variance of surface curvature) during the surface contraction wave performed on the same oocytes treated with blebbistatin, Y-27632 and ML-7 and injected with MRLC and corresponding DMSO control, confirming the effectiveness of the treatments. Dot plots of measurements on individual oocytes overlaid with box plots of the same data. N indicates the number of oocytes. ANOVA, $p < 0.0001$. Data were collected from three independent experiments.

DOI: https://doi.org/10.7554/eLife.31469.012

The following figure supplement is available for figure 5:

**Figure supplement 1.** Myosin Vb tail overexpression does not affect the rate of contraction.

DOI: https://doi.org/10.7554/eLife.31469.013

## Filament stabilization slows contraction in a dose-dependent manner consistent with a disassembly-driven mechanism

As shown above, the mass of F-actin in the old network continuously decreases, and filament stabilization inhibits contraction, suggesting a direct link between F-actin disassembly and network contraction. To test this, we titrated in a stabilizing activity by injecting increasing amounts UtrCH using a quantitative microinjection method (*Jaffe and Terasaki, 2004*). Analysis of pair-wise chromosome approach showed that UtrCH slowed chromosome transport in a dose-dependent manner (*Figure 6A,C*). We derived the contraction rates for each condition as for *Figure 1E,F*, and using the contraction rate we calculated the size of the old network and the F-actin mass contained within, as for *Figure 4A*. Analysis of *F-actin mass* in the old network revealed that UtrCH slows net F-actin disassembly in a dose-dependent manner (*Table 2*, *Figure 6B,D*). Quantification of changes in mean UtrCH intensity in a fixed size region on the other hand revealed similar *F-actin density* despite the very different contraction rates (*Figure 6E* and *Figure 4—figure supplement 1B*, respectively). In other words, stabilization by UtrCH did not render the network denser or sparser, it merely slowed its contraction. As a result, contraction rates ($\alpha_0$) and decrease in F-actin mass were reduced in a correlated manner, indicating a direct coupling between disassembly and contraction.

We next compared these quantitative observations to the predictions of models of motor- and disassembly-driven contractile mechanisms. We first determined the viscoelastic parameters from control experiments (i.e. oocytes injected with PBS; *Supplementary file 2*) that were very similar to those for untreated oocytes (*Table 1*). We then used the measured disassembly rate $k_0$ and the dispersion of filament length $\lambda$ to predict the contraction rate for each condition (without changing any other viscoelastic parameters) (*Table 2*). The two models, motor- and disassembly-driven, predicted very distinct behaviors in response to increasing UtrCH concentrations. For intermediate and high UtrCH levels, the motor-driven model predicted a fast increase of contraction rate, in contrast to the disassembly-driven model predicting contraction rate to remain nearly constant over time, well matching experimental observations (*Figure 6F*). Consistently, the initial contraction rate, $\alpha_0$ derived from the disassembly-driven, but not from the motor-driven model, predicted contraction rates in response gradual stabilization (*Table 3*).

This is explained by the fact that in the disassembly-driven mechanism contraction and disassembly rates are inherently coupled. In the motor-driven model, decreasing disassembly rate caused an increase in F-actin concentration that resulted in more effective motor action, leading to the observed increase of contraction rate over time. Similar results were obtained with a purely viscous model (*Figure 6—figure supplement 1* and *Supplementary file 1*).

## Enhanced filament disassembly speeds up contraction as predicted by the disassembly-driven model

To further test the active role of disassembly in driving contraction, we used Latrunculin A, a toxin that sequesters actin monomers. Therefore, Latrunculin A is expected to block actin filament assembly and to enhance depolymerization by reducing the effective monomer concentration (*Spector et al., 1989*). Thus, in a dynamically assembling and disassembling network the net effect will be shifting the equilibrium towards disassembly.

We observed a dramatic increase in contraction rate upon acute treatment with a high dose of Latrunculin A (*Figure 7A*, *Video 3*). Starting approx. 50 s after Latrunculin A addition, the network diameter begun to decrease at rates approx. three-fold higher than DMSO controls (*Figure 7B*). Quantification of *F-actin mass* in the rapidly contracting old network revealed that F-actin amount decreased at a correspondingly faster rate (*Figure 7C*, *Figure 4—figure supplement 1C* and *Table 2*).

Similar to above, we next determined the viscoelastic parameters of the network in the DMSO control (*Supplementary file 3*, note that this time $\alpha(t)$ was derived from fitting the network diameter since chromosomes escape and thus cannot be used for the analysis, *Figure 7—figure supplement 1*). We then used the measured disassembly rate $k_0$ and the dispersion of filament length $\lambda$ to predict $\alpha(t)$ by the two models (*Table 2*). While the motor-driven model predicted the contraction rate to change little upon increased disassembly, the disassembly-driven model predicts an increase in contraction rate closely matching experimentally observed values (*Figure 7D*).

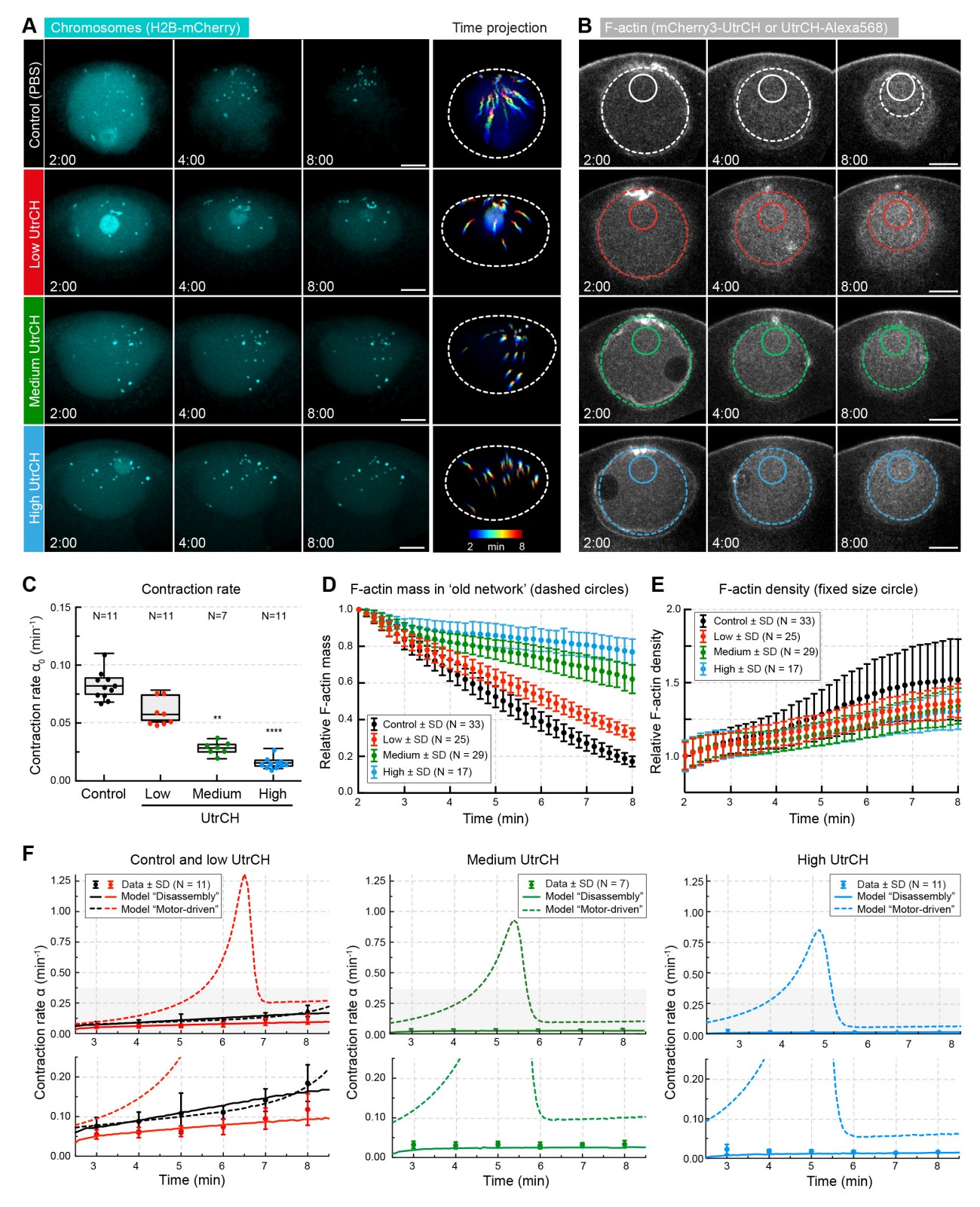

**Figure 6.** Stabilization of filaments slows disassembly and contraction rate. (**A**) Maximum-intensity z projections of the nuclear region of oocytes expressing H2B-mCherry (cyan) and injected with different amounts of recombinant UtrCH or PBS as control. Dashed circles represent the initial nuclear contour. Right: pseudocolored time projection of z-projections. (**B**) Selected confocal sections through the nuclear regions of live oocytes expressing mCherry3-UtrCH. The extrapolated size of the old network shown by colored dashed circles that were used for calculation of F-actin mass. Region of
*Figure 6 continued on next page*

*Figure 6 continued*

constant size shown by colored solid circles is used to calculate F-actin density. (C) Contraction rates ($\alpha_0$) were calculated as for *Figure 1F*. Box plots combined with dot plots. Kruskal-Wallis' post hoc test, **p<0.008; ****p<0.0001. (D) Quantification of F-actin mass in the old network in the different conditions shown in (B) and calculated as illustrated in *Figure 4A*. (E) Quantification of F-actin density in a constant size region shown in (B). (F) Fit of observed contraction rates to motor- and disassembly-driven models of contraction (dashed and continuous lines, respectively). Scale bars: 20 µm; time is given as mm:ss relative to NEBD. N indicates the number of oocytes. Data were collected from at least three independent experiments.
DOI: https://doi.org/10.7554/eLife.31469.014

The following figure supplement is available for figure 6:

**Figure supplement 1.** The rates of disassembly and contraction are coupled, and predicted by disassembly-driven model.
DOI: https://doi.org/10.7554/eLife.31469.015

In summary, the observed changes in contraction due to perturbations of motor activities, as well as stabilizing and destabilizing filaments, are well predicted by a model in which contraction is driven by disassembly, but not by a motor-driven model. While in many aspects the two models are similar (e.g. both models include the 'fluidization' of the network by disassembly), the principal difference is that in the disassembly-driven model disassembly and contraction are directly coupled.

## In silico reconstitution of the contractile system predicts a 'depolymerization harnessing factor' essential to drive contraction

Our data suggest that the F-actin network transporting chromosomes in starfish oocytes contracts by a mechanism mediated by filament disassembly. To explore the mechanism at the filament level, we implemented an agent-based model in the Cytosim software (*Nedelec and Foethke, 2007*). We based the model on recently published cytosim simulations, used to support a general theoretical framework of contractility in disordered filament networks, albeit in that case with static filaments (*Belmonte et al., 2017*). As in the published model, we used a circular 2D geometry, assuming a 'nuclear region' with a radius of 10 µm initially filled by filaments having the rigidity of F-actin. We then added a few features to model the starfish chromosome transport system: first, disassembly occurring at the same rate for all filaments throughout the network; and second new filament assembly at the boundary. We also introduced 'chromosomes', passive objects embedded in the network for the purpose of producing simulated trajectories.

We first tested whether the combination of disassembly and cross-linking could drive network contraction. Consistent with previous theory and experiments (*Belmonte et al., 2017*; *Bendix et al., 2008*; *Braun et al., 2016*), in the absence of molecular motors the disassembling network did not contract over a broad range of parameters (number of filaments, cross-linkers and their unbinding rates) tested (*Figure 8—figure supplement 1A–C*).

This suggested that contraction in our system either requires a molecular motor or a specific molecular activity that is capable of harnessing F-actin disassembly to produce contractile force. This latter can be a molecule capable of tracking the depolymerizing end of a filament and forming a connection to a neighboring filament at the same time (*Zumdieck et al., 2007*). Introducing such elements into the simulations, referred to as end-binding cross-linkers, resulted in robust large-scale contraction of the disassembling network in the absence of molecular motors (*Figure 8—figure supplement 1A,B*).

We then tested the effects of assembly of new filaments at the network boundary. This reproduced well the observed behavior, whereby newly produced filaments fill up the space left by the 'old' contracting network. Interestingly, we found that contraction rate is rather insensitive to the rate of filament assembly at the boundary: insufficient or excess filament production rates led to inhomogeneity in the network, but this did not significantly affect the rate of contraction (*Figure 8—figure supplement 1D*). This suggests that the contraction of the old network, and new filament assembly at the NER boundary do not necessarily need to be coupled, and this is consistent with experimental observations. First, we observed an accumulation of filaments at the boundary of old and new networks in oocytes in which contraction and filament production was imbalanced by UtrCH stabilization (*Figure 8—figure supplement 1E*). Second, in untreated oocytes we observed a gradual accumulation of filaments at the NER boundary from the vegetal to the animal pole. This can be explained by constant rate of filament production all along the boundary, and while these produced filaments are distributed to fill in the relatively larger gap at the vegetal pole, filaments accumulate

**Table 2.** Disassembly rates $k_0$ (min$^{-1}$) and dispersion of F-actin length distributions.

| | UtrCH-stabilized | | | | Lat A-treated* | | SMIFH2-treated | |
|---|---|---|---|---|---|---|---|---|
| | Control | Low | Medium | High | Control | Lat A | Control | SMIFH2 |
| $k_0$ | 0.162 | 0.126 | 0.073 | 0.048 | 0.0819 | 0.5106 | 0.1995 | 0.1650 |
| $\lambda$ | 0.677 | 0.889 | 6.946 | 709.4 | 0.0671 | 14.66 | 1.555 | 1.540 |

*The solvent, DMSO had an effect on the viscolelastic parameters even in controls (**Supplementary file 3**).

DOI: https://doi.org/10.7554/eLife.31469.016

at the animal pole where there is no gap (**Figure 8—figure supplement 1E**). Thus it appears that filament production does not contribute to force production; it rather serves to fill in the gap between the 'old network' and cytoplasmic F-actin networks, mechanically isolating the contractile system in the nuclear region from the cytoplasm (**Figure 8—figure supplement 1E**).

As the final step, we combined these components of the contractile system, and set the rate of global disassembly and assembly at the boundary to match experimental observations. We then simulated two scenarios: (i) contraction driven by an end-binding cross-linker harnessing depolymerization for generating contractile force; (ii) myosin II-like motor activity driving contraction of the disassembling network (in presence of cross-linkers, but without end-binding cross-linkers) (**Figure 8A,B**). Both simulations produced robust contraction. However, while contraction mediated by the end-binding cross-linker closely reproduced several key features of the experimental system, motor-driven contraction showed a distinct behavior not observed experimentally (**Figure 8C**, **Video 4**). First, the disassembly-driven mechanism invariably produces contraction with a constant speed, as observed experimentally, while motor-driven contraction typically accelerates (**Figure 8C**). Second, while a strictly quantitative comparison of the 2D simulations and the 3D experimental system is not possible, network density remains nearly constant during contraction in the disassembly-driven simulations, as in experiments, while during motor-driven contraction network density continuously increases (**Figure 8D**). Further, the simulations reproduce the fact that contraction rate for a disassembly-driven mechanism is tightly coupled to the rate of disassembly (**Figure 8E**).

Taken together, the simulations are consistent with the active gel theory, and additionally identify the 'disassembly harnessing factor' as the key component driving the contractile system. By searching the literature for molecular candidates for such factor with features predicted by the simulations, we found that a formin-like molecule may fit these requirements. It has been shown that in vitro and under specific conditions the formin, mDia1 is able to track depolymerizing ends and generate pulling forces on the disassembling filament end (**Jégou et al., 2013**; **Mizuno et al., 2007**). In addition, formins have been shown to have cross-linking activities on their own or by binding to other actin regulators such as Spire (**Esue et al., 2008**; **Montaville et al., 2014**). In this way a hypothetical formin may serve as an effective 'harnessing factor'.

We tested this hypothesis by inhibiting all formin-like activities in the oocyte by SMIFH2, a small-molecule inhibitor of the FH2 domain contained and essential for the activity of all known formins (**Rizvi et al., 2009**), and which has already been evaluated in mouse and starfish oocytes (**Kim et al., 2015**; **Ucar et al., 2013**). We thus quantified the contraction rate and F-actin mass in oocytes treated with SMIFH2 and compared to DMSO controls (**Figure 9A–D** and **Figure 4—figure supplement 1D**, **Table 2**). Impairing formin activity led to a significant decrease of network contraction rate and a reduction of F-actin mass loss, similar to stabilization by UtrCH, and well predicted by the disassembly-driven model (**Figure 9E**; **Table 4** and **Supplementary file 4**).

While these results suggest that a formin-like molecular activity may be harnessing the free energy originating from disassembly of actin filaments, unfortunately the experimental tools currently available in starfish oocytes are limited, and thus identification of this factor will remain a challenge for the future.

## Discussion

Animal oocytes typically have an exceptionally large nucleus storing nuclear proteins such as histones needed for early embryonic development. Oocytes also divide very asymmetrically in order to retain these nutrients in a single egg. The requirement for this highly asymmetric division constrains the

**Table 3.** Contraction rates $\alpha_0$ (min$^{-1}$ ± S.D.) in response to increasing stabilization (UtrCH).

| | UtrCH injection | | | |
| --- | --- | --- | --- | --- |
| | Control | Low | Medium | High |
| Experiment | 0.0831 ± 0.012 | 0.0612 ± 0.011 | 0.0280 ± 0.006 | 0.0164 ± 0.005 |
| Model M | 0.0818 | 0.1821 | 0.2063 | 0.2133 |
| Model D | 0.0825 | 0.0586 | 0.0226 | 0.0117 |

DOI: https://doi.org/10.7554/eLife.31469.017

size of the meiotic spindle. Therefore, across animal species, the meiotic spindle is small and this necessitates additional mechanism to collect and transport chromosomes scattered in the large nuclear volume to the forming microtubule spindle.

Here, we propose and reconstruct in silico a simple and robust architecture for an F-actin-based contractile system adapted to the essential function of transporting chromosomes in the large oocyte. First, the process is triggered by NEBD opening the previously inaccessible nuclear compartment to cytoplasmic actin, actin assembly factors and cross-linkers resulting in the rapid assembly of a cross-linked F-actin network in the nuclear region. As revealed by our simulations, the only additional components necessary are: (i) filament production along the boundary to separate the nuclear region from cytoplasmic F-actin networks allowing contraction towards the center, and (ii) overall filament dynamics shifted towards disassembly. In the presence of a 'depolymerization harnessing factor' this system undergoes robust contraction exhibiting features that closely match the experimental observations. These include constant speed, stable network density during contraction (well matched to the size of chromosomes [*Mori et al., 2011*]), and graded response to filament stabilization and destabilization, but no effect of motor inhibition. We find that the rate of disassembly is the key parameter controlling the rate of contraction, while contraction rate is broadly insensitive to filament production at the boundary. Finally, an additional elegant feature of the system is that the disassembling network will eventually release the transported chromosomes, as it loses connectivity and stops contracting. By contrast, in the absence of disassembly, a contracting network would necessarily densify, in which case the chromosomes would be surrounded by a tight meshwork of F-actin at the end of contraction, which would likely interfere with their capture by spindle microtubules.

In conclusion, we investigated here a contractile system adapted to transport chromosomes in large oocytes. Both experiments and simulations point to an F-actin disassembly-driven mechanism driving this process, as a simple and robust solution to the challenge posed by the size and geometry of the starfish oocyte. Network contraction can be seen here as a large-scale treadmilling system, where F-actin disassembles in bulk and is replenished by filaments polymerized at the NER boundary. However, the evidences lead us to conclude that de-novo filament assembly at the NER boundary does not participate in force generation, which rather originates in the bulk of the network.

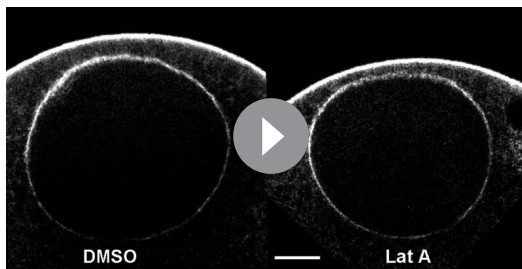

**Video 3.** Contraction of the F-actin network accelerates upon acute treatment with Latrucunlin A. Oocytes expressing 3mEGFP-UtrCH to visualize F-actin were acutely treated either with DMSO (left) or Latrunculin A (right). Scale bar: 20 μm.
DOI: https://doi.org/10.7554/eLife.31469.020

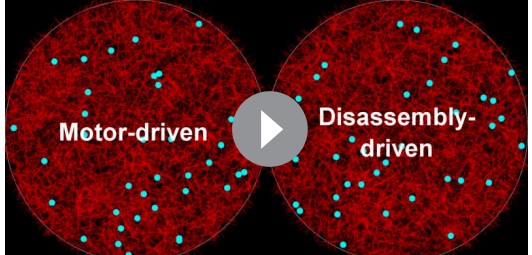

**Video 4.** Cytosim simulations of motor- and disassembly-driven mechanisms to generate contraction. Simulations show production of filaments (gray) while initially present filaments (red) are contracting to transport chromosomes (cyan).
DOI: https://doi.org/10.7554/eLife.31469.023

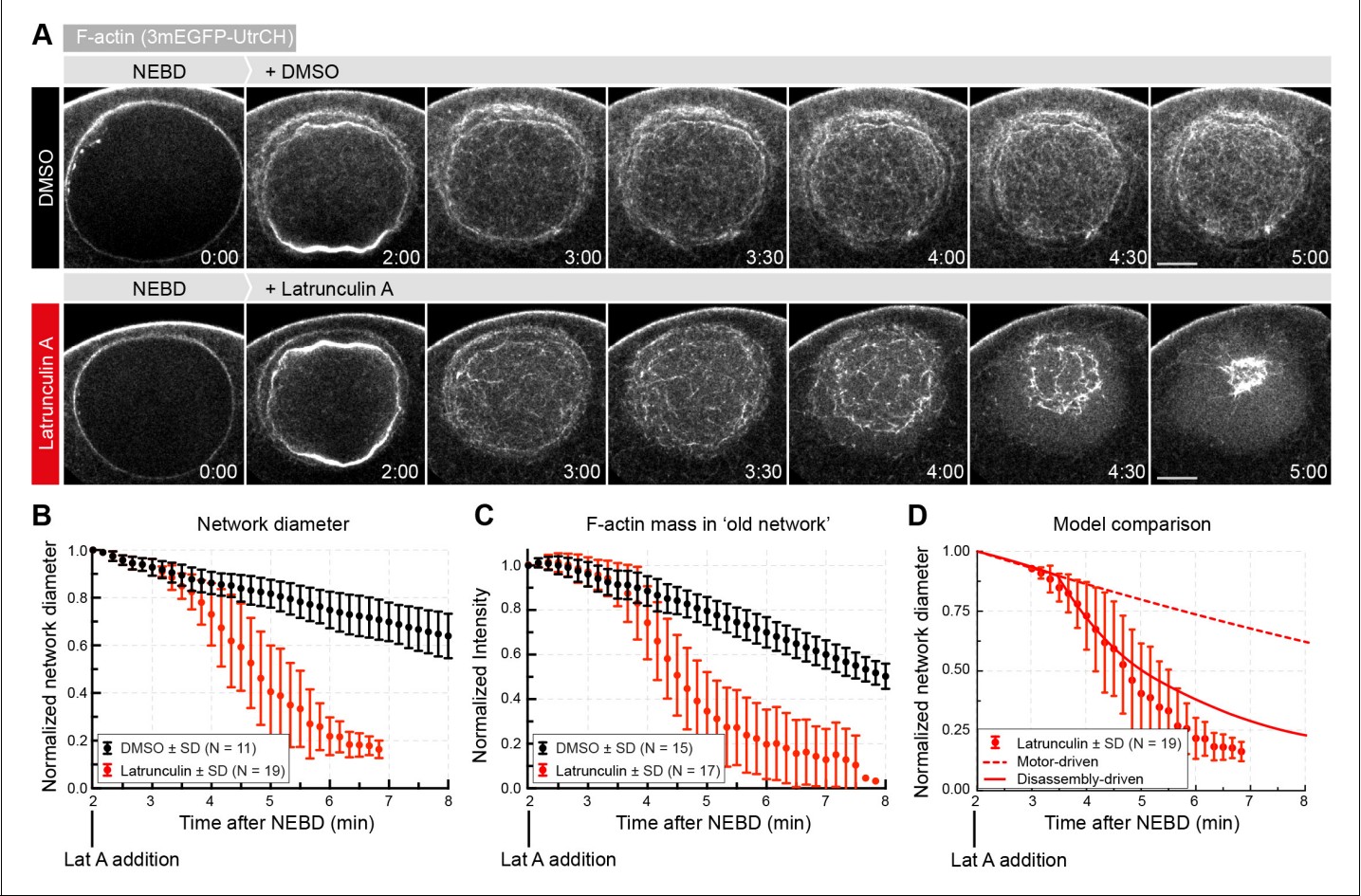

**Figure 7.** Enhancing disassembly speeds up contraction. (**A**) Selected frames from a time lapse of confocal sections through the nuclear region of live oocytes expressing mEGFP3-UtrCH (gray). Oocytes were treated with either Latrunculin A (2.5 µM) or a corresponding amount of DMSO ~2 min after the start of NEBD. (**B**) The size of the contracting network was calculated similar to *Figure 2D*. (**C**) Quantification of F-actin mass in the old network as described in *Figure 4A*. (**D**) Comparison of fits of motor- and disassembly-driven models to the observed network contraction rate (dashed and continuous lines, respectively). N indicates the number of oocytes. Data were collected from at least three independent experiments.
DOI: https://doi.org/10.7554/eLife.31469.018

The following figure supplement is available for figure 7:

**Figure supplement 1.** Enhancing filament disassembly by Latrunculin A leads to chromosome loss during contraction.
DOI: https://doi.org/10.7554/eLife.31469.019

Key questions remain open regarding the detailed molecular mechanism underlying disassembly-driven force generation. The first question is the exact source of free energy driving the process. While actin polymerization is well established to produce force, at cell protrusions, for example (*Footer et al., 2007*; *Krause and Gautreau, 2014*), force production by disassembly has so far only been evidenced in vitro for specific actin concentration (*Jégou et al., 2013*). It is clear that microtubule depolymerization can produce force (*Grishchuk et al., 2005*), that is harnessed for example to drive separation of sister chromatids in anaphase (*Cheeseman and Desai, 2008*; *Westermann et al., 2006*). Similar to tubulin, actin polymerization is tightly coupled to nucleotide hydrolysis driving the system out of equilibrium that may enable generation of force by depolymerization (*Braun et al., 2016*). Alternatively, entropic forces generated either by crowding agents surrounding filaments or by diffusible cross-linkers confined in filament overlaps could drive contraction (*Braun et al., 2016*), but we have not tested this possibility.

Another important question is the identity and nature of the factor harnessing the free energy originating from disassembly of actin filaments, and converting it into contractile force. Intriguingly,

experiments in vitro and our initial experiments in starfish oocytes hint at formins as potential candidates for this function. These results need to be confirmed, since they rely on a small-molecule inhibitor, SMIFH2, which has been shown to inhibit formins in starfish oocytes (*Ucar et al., 2013*), but its specificity and selectivity profile is yet to be characterized. Secondly, the observed effect may be indirect resulting from the altered balance between other formin-driven actin assembly pathways and Arp2/3 nucleated structures, as recently showed in yeast (*Burke et al., 2014*). Finally, although Jégou and colleagues clearly demonstrated tracking of filament ends by mDia1 (*Jégou et al., 2013*), we are at present unable to assess how similar or different these in vitro conditions may be from those in starfish oocytes, where (dis)assembly dynamics at filaments ends is likely to be regulated by multiple, so far unidentified factors.

It will be very exciting to test in the future whether formins, so far primarily considered to promote polymerization, may be also involved in transmitting force at disassembling filaments ends in vivo. However, at this point we are not able to exclude other scenarios, such as rapid re-binding of a cross-linker to the shrinking filament end (*Mendes Pinto et al., 2012*). Further, the rate of disassembly could be affected by severing, generating more depolymerizing ends, and the actual rate of depolymerization at the end set by capping and depolymerizing factors. More elaborate models are necessary to explore these scenarios. In either case, the contractile 3D network we analyzed here, adapted to the task of transporting chromosomes in the large starfish oocyte, will continue to be a valuable system to further explore the important mechanistic details of the so far poorly understood motor-independent mechanisms of contraction.

# Materials and methods

## Key resources table

| Reagent type or resource | Designation | Source or reference |
|---|---|---|
| Biological sample | | |
| Patiria miniata | Patiria miniata | https://scbiomarine.com/ |
| Transfected construct | | |
| MRLC (Patiria miniata) | MRLC-mEGFP | doi:10.1038/s41467-017-00979-6 |
| H2B (human) | H2B-mCherry, H2B-3mEGFP | doi:10.1038/nmeth876 |
| Utrophin CH domain (human) | mEGFP3-UtrCH, 3mCherry-UtrCH | doi:10.1002/cm.20226 |
| myosinVb tail domain (mouse) | myosinVb-Tail | doi:10.1038/ncb2802 |
| Peptide, recombinant protein | | |
| Histone H1 (calf) | H1 | Merck |
| Utrophin CH domain (human) | UtrCH | doi:10.1002/cm.20226 |
| Commercial assay or kit | | |
| AmpliCap-Max T7 High Yield Message Maker | AmpliCap-Max T7 High Yield Message Maker | CellScript |
| Poly(A) tailing kit | Poly(A) tailing kit | CellScript |
| Gel filtration column PD-10 | Gel filtration | GE Healthcare |
| Ni-NTA resin | Ni-NTA resin | Qiagen |
| Vivaspin column 10,000 MW | Vivaspin column | Sartorius |
| Alexa Fluor 568 succinimidyl ester | Alexa Fluor 568 succinimidyl ester | Invitrogen |
| Alexa Fluor 488 succinimidyl ester | Alexa Fluor 488 succinimidyl ester | Invitrogen |
| Chemicals, drugs | | |
| DiIC$_{16}$(3) | DiI | Invitrogen |
| 1-methyladenine | 1-MA | ACROS organics |
| Phalloidin-AlexaFluor 568 | Phalloidin-AlexaFluor 568 | Invitrogen |
| LatrunculinA | Lat A | Abcam |
| SMIFH2 | SMIFH2 | Tocris |

*Continued on next page*

*Continued*

| Reagent type or resource | Designation | Source or reference |
|---|---|---|
| Y-27632 | Y-27632 | Enzo Life Sciences |
| ML-7 | ML-7 | Tocris |
| Blebbistatin | BB | Abcam |
| Software, algorithm | | |
| Matlab | Matlab | Mathworks |
| Cytosim | Cytosim | doi:10.1088/1367-2630/9/11/427 |

## Oocyte injection, maturation and centrifugation

Starfish (*P. miniata*, also known as *Asterina miniata*) were obtained from Southern California Sea Urchin Co., Marinus Scientific, South Coast Bio-Marine, or Monterey Abalone Co. and maintained in seawater tanks at 15°C at the European Molecular Biology Laboratory (EMBL) Marine Facility. Oocytes were isolated and injected using mercury-filled needles as described elsewhere (*Jaffe and Terasaki, 2004*). Recombinant protein markers were injected shortly (10–15 min) after initiation of meiosis, whereas mRNA injections were done the day before and incubated overnight at 14°C to obtain sufficient levels of protein expression. Meiotic maturation was triggered by the addition of 10 μM 1-methyladenine (ACROS Organics). NEBD typically started 25 min after hormone addition, and only oocytes starting NEBD between 15 and 35 min were analyzed. Oocyte centrifugation was performed at 2500 rpm for 20 min (Multifuge 3; Heraeus) at 4°C, as detailed elsewhere (*Matsuura and Chiba, 2004*; *Mori et al., 2011*).

## Live cell fluorescent markers

H2B-3mEGFP, H2B-mCherry (*Neumann et al., 2006*), mEGFP3- and mCherry3-UtrCH (*Burkel et al., 2007*) and MRLC-mEGFP (*Bischof et al., 2017*) were subcloned into pGEMHE for in vitro transcription as described elsewhere (*Lénárt et al., 2003*). Mouse sequence of Myosin Vb Tail was used to search for homologs in the *P. miniata* transcriptome by BLASTP (*Altschul et al., 1990*). Hits were considered homologs when the e-value was $< 10^{-20}$. Identified cDNA sequence was synthetized by GENEWIZ, fused to mEGFP and subcloned into pGEMHE for in vitro transcription. Capped mRNAs were synthesized from linearized templates using the AmpliCap-Max T7 High Yield Message Maker and extended with poly(A) tails using the Poly(A) Tailing kit (CellScript). mRNAs were dissolved in 11 μl RNase-free water (typically at 8–12 μg/μl) and injected to 1–5% of the oocyte volume. Histone H1 from calf thymus (Merck) was labeled with AlexaFluor 568 succinimidyl ester (Invitrogen) according to the manufacturer's instructions, purified, concentrated by Vivaspin columns (10,000 MW, Sartorius) and was injected to 0.1% of the oocyte volume. DiIC$_{16}$(3) (Invitrogen) was dissolved in vegetable oil to saturation and injected into oocytes as in (*Lénárt et al., 2003*). The methanol stock of phalloidin-AlexaFluor 568 (Invitrogen) was dried and dissolved in PBS (pH 7.2) and amounts corresponding to $10^{-5}$ units were injected directly into the nuclear region of oocytes ~ 1 min after NEBD. Utrophin CH domain (UtrCH) (*Burkel et al., 2007*) was subcloned into pET24d(+) for expression in *E. coli*. The recombinant protein was expressed for 1 hr at 37°C and purified on a Ni-NTA resin (QIAGEN). The purified protein was labeled by AlexaFluor 488 and AlexaFluor 568 succinimidyl ester (Invitrogen) according to the manufacturer's instruction, purified by gel filtration (PD-10, GE Healthcare) and concentrated by Vivaspin columns (10,000 MW, Sartorius) before injection to oocytes. Fluorescently labeled recombinant UtrCH was injected 5–10 min prior to NEBD.

## Drug treatments

For drug treatments, oocytes were transferred into plastic dishes (Ibidi #80131). To inhibit myosin II ATPase activity, oocytes were incubated with a final concentration of 300 μM Blebbistatin (Abcam) for at least 3 hr before starting maturation; with ML-7 (Tocris) and Y-27632 (Enzo Life Sciences) at a final concentration of 100 μM for 1 hr. To induce acute F-actin depolymerization, oocytes were first matured and then treated with Latrunculin A (Abcam) after NEBD, as NEBD is known to depend on actin polymerization in starfish oocytes (*Mori et al., 2014*). To inhibit formin FH2 activity, we incubated oocytes with SMIFH2 (Tocris) at a final concentration of 50 μM for at least 2 hr.

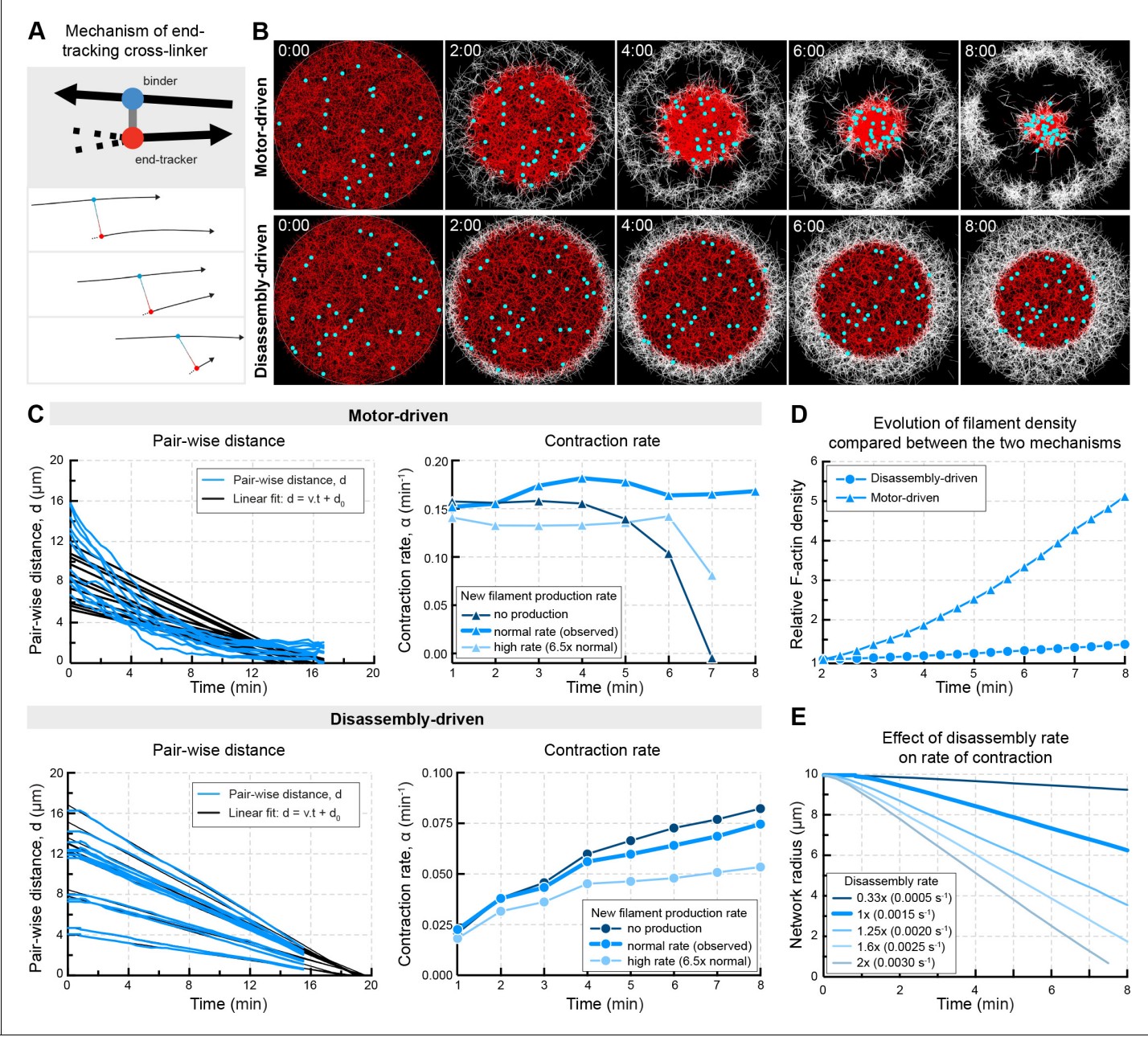

**Figure 8.** In silico reconstruction of a disassembly-driven contractile system. (**A**) Top: schematic representation of the mechanism by a hypothetical end-tracking cross-linker serving as 'depolymerization harnessing factor'. Bottom: zoom on two filaments linked by such end-tracking cross-linker as implemented in simulations in (**B**). The blue 'head' corresponds to a binder subunit, the red head binds and tracks the depolymerizing end. (**B**) Snapshots of 2D Cytosim simulations for motor- and disassembly-driven mechanisms. Simulations start with 5000 filaments (red) of a length of 1.5 μm and a disassembly rate of 0.0015 μm/s for all filaments. New filaments (white) are added at the boundary at a constant rate of 12 filaments per second. Chromosomes attached to the network are shown in cyan. (**C**) Left panels: plot of pair-wise distances of chromosomes, d versus time derived from simulations in (**B**). Right panels: Contraction rates calculated for 2 min intervals, similar to *Figure 1E,F*, for different rates of filament production. (**D**) Filament density derived from the simulations shown in (**A**). (**E**) The radius of the contracting 'old' network was extracted from simulations similar to that shown in (**A**) testing the effect of disassembly rates in the disassembly-driven mechanism.

DOI: https://doi.org/10.7554/eLife.31469.021

The following figure supplement is available for figure 8:

**Figure supplement 1.** Network contraction is independent of the rate of filament production at the boundary.
DOI: https://doi.org/10.7554/eLife.31469.022

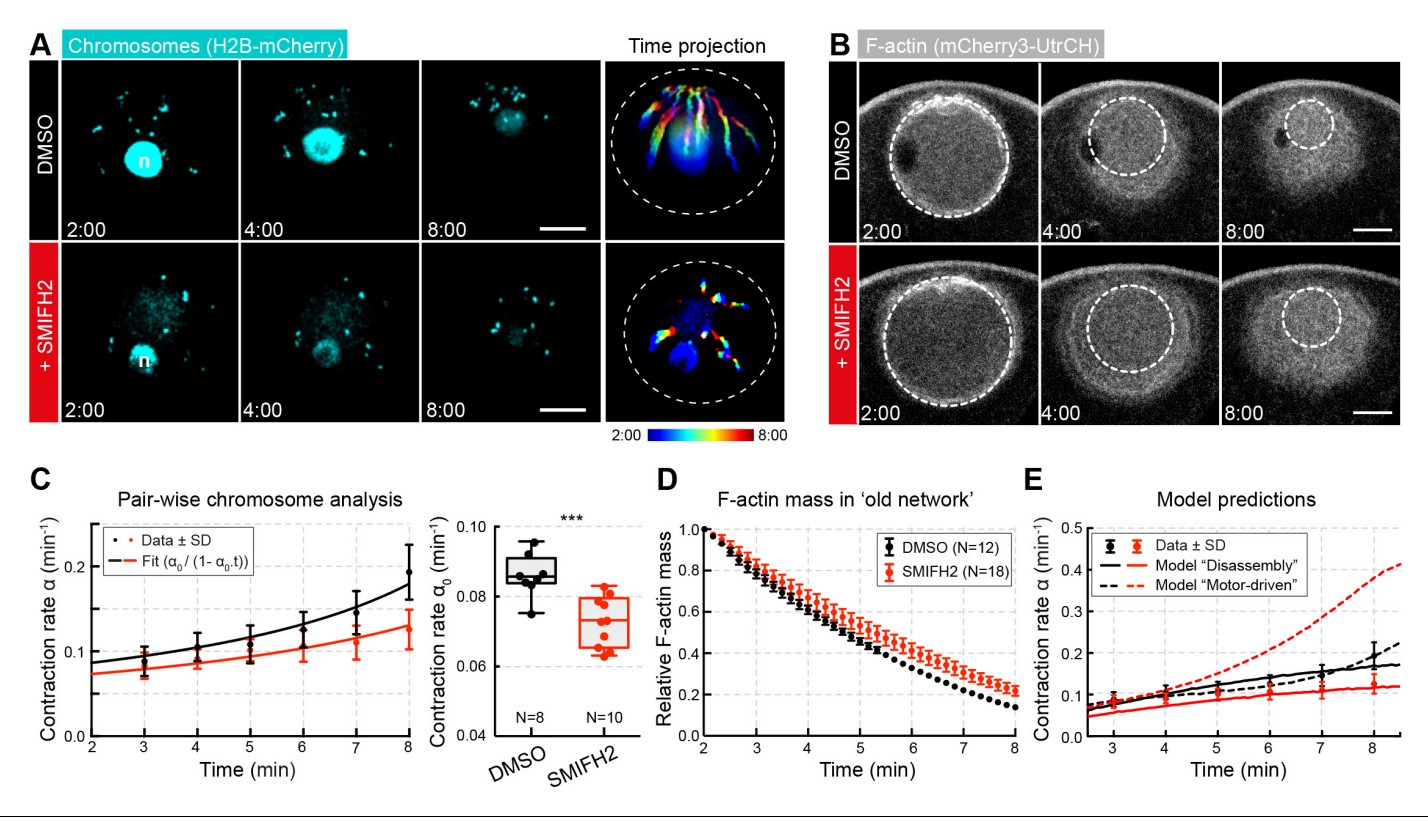

**Figure 9.** Inhibiting formin FH2 domain activity slows contraction. (A) Maximum-intensity z-projection through the nuclear region of oocytes expressing H2B-mCherry (cyan), either incubated with DMSO or SMIFH2 (50 μM). Right: pseudocolored time projection of z-projections. Dashed circles outline the initial nuclear contour. Scale bar: 20 μm; time is given as mm:ss relative to NEBD. (B) Selected confocal sections through the nuclear regions of live oocytes expressing mCherry3-UtrCH. The extrapolated size of the old network shown by dashed circles is used for calculation of F-actin mass. (C) Left: contraction rate over time for each condition with fits for determining $\alpha_0$ as in *Figure 1F*. Right: box plots combined with dot plots of $\alpha_0$ for multiple oocytes. N indicates the number of oocytes. Mann-Whitney's test, ***$p<0.0005$. Data collected from three independent experiments. (D) Quantification of F-actin mass in the old network calculated as in *Figure 4A*. (E) Fit of observed contraction rates to motor- and disassembly-driven models of contraction (dashed and continuous lines, respectively).

DOI: https://doi.org/10.7554/eLife.31469.024

## Light microscopy and laser ablation

Microscopy was done on a Leica SP5 or SP8 confocal microscope equipped with a fast Z-focusing device (SuperZ galvo stage) and using a 40x HCX PL AP 1.10 NA water immersion objective lens (Leica Microsystems). To record chromosome transport, starfish oocytes were imaged in 3D (a Z-step of 2 μm over 70 μm) over time (time step of 5 s) using a square frame of 256 × 256 pixels at a pixel size of 447 nm. To monitor F-actin network contraction, starfish oocytes were imaged in 3D centered on the median plane (plane of biggest surface) with a Z-step of 2 μm covering 10–20 μm over time (time step 10 s) using a square frame of 512 × 512 pixels at a pixel size of 223 nm. All imaging was performed at room temperature (20–22°C).

A pulsed two-photon laser (Chameleon, Coherent) was interfaced to a LSM780 confocal microscope (Zeiss) through a customized optical path aligned with the optical axis of the microscope. We sequentially imaged (a Z-step of 2 μm over 20 μm every 10 s) and performed 3D ablation (typical duration ~ 1 s) by imaging a stack of 1 × 50×60 μm, with a Z-step of 2 μm at 800 nm (30% laser power). Imaging was started ~ 1 min before the 3D ablation.

## Image processing and data analysis

Image processing was done using Fiji (*Schindelin et al., 2012*). Figures were assembled in Adobe Illustrator. Data analyses were performed using Matlab (MathWorks). Image data are shown after

brightness and contrast adjustment and application of a 2D Gaussian blur filter (sigma value: 0.2–1.0). Some image data were registered using the 'StackReg' plugin implemented in Fiji to correct for oocyte movements during experiments. Panels show either single Z-slices, maximum or sum intensity projections as indicated in the figure legends. Colored time projections were done using the Time-RGBcolorcode macro in Fiji. Chromosome tracking and data analysis were done by in-house routines written in Matlab. Segmentation and tracking algorithms are described elsewhere (*Monnier et al., 2012*). Pair-wise relative chromosome velocities were defined as the slope of the best-fit line ($R^2 > 0.9$) to the separation distance between the pair versus time. Similarly, pair-wise chromosome approach rate was defined as the slope of the linear fit to the separation distance between the pair for every 2 min interval during the actin-driven transport phase (from 2 to 8 min after NEBD).

To control for photobleaching, we measured fluorescence intensity in a small region in the cytoplasm during the time course of the experiment. Oocytes in which the fluorescence intensity decrease exceeded 10%, were not included in further analysis.

To measure the mass of F-actin, a custom-written Fiji macro was first used to automatically segment the nuclear region based on either the $DiIC_{16}(3)$ fluorescence or the bright field channel. This pipeline included a FeatureJ Hessian filtering, followed by a 2D Gaussian filtering and automatic thresholding using the 'Otsu' or 'Huang' algorithm. The segmented area was then used on the F-actin channel to measure the density ('mean') (*Figure 4A* and *Figure 4—figure supplement 1*). To estimate the mass of F-actin within the contracting network over time, we used the initial outline of the nucleus at the onset of the actin-driven phase, and applied iteratively the function 'Enlarge' with a factor equals to $-(\alpha_0)$ to extrapolate the area and further the volume of the contracting network. Thus, the mass over time was measured as the product of the density and the extrapolated volume at each time point. For Latrunculin A-treated oocytes, we proceeded in two steps: before the addition of Latrunculin A, we extrapolated the size of the old network using $\alpha_0$ derived from chromosome tracking. After drug addition, we measured the size of contracting old network, and derived the contraction rate similar to *Figure 2E*.

Surface curvature measurements were performed using an algorithm written in Matlab and described elsewhere (*Bischof et al., 2017*). The strength of surface contraction waves was defined by the maximum variance of the radius of curvature. The Matlab script is available in the *Supplementary file 6*. Statistical tests used were unpaired nonparametric tests (Mann-Whitney and Kruskal-Wallis) implemented in Matlab.

## Computer simulations

The contractile networks were simulated with Cytosim (*Nedelec and Foethke, 2007*) using a Langevin dynamics approach. Actin filaments are represented by incompressible bendable filaments of rigidity 0.075 $pN.\mu m^2$ (persistence length of 18 μm) in a medium of viscosity 0.3 Pa.s (*Belmonte et al., 2017*; *Gittes et al., 1993*). Filaments have an initial length of 1.5 μm and are discretized into segments of 100 nm. Disassembly occurs from one end at a speed of 0.0015 μm/s. The other end is static. Cross-linkers are modeled as Hookean springs of 10 nm resting length and rigidity of 250 pN/μm with a binding rate $k_{on}$ = 10 $s^{-1}$ (binding range of 0.01 μm) and an unbinding rate $k_{off}$ = 0.1 $s^{-1}$ that is independent of the force exerted by the link. Motors are also Hookean springs of similar characteristics as cross-linkers, but motors additionally move on filaments with a speed of 0.02 μm/s, with a linear force-velocity relationship characterized by a maximal stall force of 6 pN. Chromosomes are modeled as point-like particles to trace the network evolution, but without a drag they do not affect its behavior. Chromosomes can bind a single filament with a binding rate $k_{on}$ = 10 $s^{-1}$ (range of 0.1 μm) and a force-independent unbinding rate $k_{off}$ = 0.0001 $s^{-1}$, and are also able to

**Table 4.** Contraction rates $\alpha_0$ ($min^{-1}$ ± S.D.) in response to SMIFH2 treatment.

|  | Control | SMIFH2 |
| --- | --- | --- |
| Experiment | 0.086 ± 0.006 | 0.073 ± 0.007 |
| Model M | 0.0859 | 0.1139 |
| Model D | 0.0848 | 0.0675 |

DOI: https://doi.org/10.7554/eLife.31469.025

track the depolymerizing end of filaments. Actin disassembly, cross-linker and motor binding and unbinding events were modeled as first-order stochastic processes (*Gillespie, 1977*).

To limit the computational cost and focus on the effect of the mode of contraction, we limited the simulation to a circular two-dimensional system of constant radius 10 μm and we neglected the steric interaction between filaments. The disc was populated initially by 5000 filaments of 1.5 μm long, 20,000 cross-linkers and 20,000 motors for the motor-driven mechanism. For the disassembly-driven mechanism, we added 40,000 end-tracking cross-linkers to the simulated system. All filaments disassembled at the same rate from one end while the other end remained stable. The disassembly rate was set at 0.0015 μm/s to match the experimentally measured contractile characteristics. To simulate the polymerization of newly-assembled filaments at the edge of the nuclear region, filaments were added with an initial length of 1.5 μm at the boundary at a rate of 12 filaments per second. Because there is no edge or steric interactions, adding new filaments per se does not produce compressive force on the old network. Cytosim is an Open Source project hosted on https://github.com/nedelec/cytosim. The configuration file is provided as *Supplementary file 1*.

## Supplementary material

The Supplementary Material includes 7 figure supplements, 4 Supplementary Movies, 6 Supplementary Files including the detailed description of the viscoelastic model (*Supplementary file 1*), 3 tables (*Supplementary file 2–4*) and the configuration file for the Cytosim model (*Supplementary file 5*) to be run in Cytosim available at https://github.com/nedelec/cytosim (*Nédélec, 2018*; copy archived at https://github.com/elifesciences-publications/cytosim). The Matlab scripts used to analyze chromosome tracks is also available (*Supplementary file 6*).

## Acknowledgements

We would like to acknowledge members of the Lenart lab for reagents and support, Johanna Bischof in particular for sharing plasmid constructs. We would like to additionally acknowledge the essential support of EMBL's Advanced Light Microscopy Facility, Protein Expression and Purification Facility, IT support, and Laboratory Animal Resources, Kresimir Crnokic in particular. The work was supported by the European Molecular Biology Laboratory, and specifically by the EIPOD Interdisciplinary Postdoctoral Programme to PB and JB, co-funded by the Marie-Curie Actions of the European Commission. SD was in part supported by the Center for Modeling in Biosciences (BIOMS).

## Additional information

### Funding

| Funder | Grant reference number | Author |
|---|---|---|
| European Molecular Biology Laboratory | | Serge Dmitrieff<br>François J Nédélec<br>Péter Lénárt |
| EU Marie Curie Actions | Cofund grant | Philippe Bun<br>Julio M Belmonte |
| Center for Modelling and Simulation in the Biosciences | | Serge Dmitrieff |

The funders had no role in study design, data collection and interpretation, or the decision to submit the work for publication.

### Author contributions

Philippe Bun, Conceptualization, Formal analysis, Supervision, Funding acquisition, Validation, Investigation, Writing—original draft, Project administration, Writing—review and editing; Serge Dmitrieff, Conceptualization, Software, Formal analysis, Funding acquisition, Validation, Investigation, Methodology, Writing—original draft, Writing—review and editing; Julio M Belmonte, Software, Formal analysis, Visualization, Methodology; François J Nédélec, Software, Formal analysis, Visualization, Project administration, Writing—review and editing; Péter Lénárt, Conceptualization,

Supervision, Funding acquisition, Writing—original draft, Project administration, Writing—review and editing

## Author ORCIDs
Philippe Bun ![ORCID] https://orcid.org/0000-0002-7975-1768
Serge Dmitrieff ![ORCID] https://orcid.org/0000-0002-1362-5670
Julio M Belmonte ![ORCID] http://orcid.org/0000-0002-4315-9631
François J Nédélec ![ORCID] https://orcid.org/0000-0002-8141-5288
Péter Lénárt ![ORCID] https://orcid.org/0000-0002-3927-248X

## Decision letter and Author response
Decision letter https://doi.org/10.7554/eLife.31469.034
Author response https://doi.org/10.7554/eLife.31469.035

## Additional files
### Supplementary files
• Supplementary file 1. Theory of the viscoelastic gel model for F-actin network.
DOI: https://doi.org/10.7554/eLife.31469.026

• Supplementary file 2. Dimensionless viscoelastic parameters for UtrCH injections.
DOI: https://doi.org/10.7554/eLife.31469.027

• Supplementary file 3. Dimensionless viscoelastic parameters for Latrunculin A treatments.
DOI: https://doi.org/10.7554/eLife.31469.028

• Supplementary file 4. Dimensionless viscoelastic parameters for SMIFH2 treatments.
DOI: https://doi.org/10.7554/eLife.31469.029

• Supplementary file 5. Cytosim configuration file to simulate a contracting F-actin network.
DOI: https://doi.org/10.7554/eLife.31469.030

• Supplementary file 6. Matlab scripts used to analyze chromosome tracks.
DOI: https://doi.org/10.7554/eLife.31469.031

• Transparent reporting form
DOI: https://doi.org/10.7554/eLife.31469.032

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
