## [Decision Letter]

Thank you for submitting your article "Disassembly-driven contraction of an F-actin network transports chromosomes in starfish oocytes" for consideration by *eLife*. Your article has been reviewed by three peer reviewers, one of whom, Pekka Lappalainen (Reviewer #1), is a member of our Board of Reviewing Editors, and the evaluation has been overseen by Andrea Musacchio as the Senior Editor.

The reviewers have discussed the reviews with one another and the Reviewing Editor has drafted this decision to help you prepare a revised submission.

Summary:

Contractile actomyosin structures are critical for many morphogenetic events and transport processes in cells. Recent studies demonstrated that a three-dimensional contractile actin network collects chromosomes and transports them to the forming microtubule spindle in starfish oocytes. However, the mechanism by which this network contracts has remained elusive. Here, Bun et al., provide evidence that myosin activity is not necessary for the network contraction, but that actin filament disassembly instead appears to drive the network contractility.

All reviewers found this study very interesting and stated that the experimental part is novel and convincing. However, because the experiments are not sufficiently well connected to the model (and the model was not rigorously tested experimentally), we cannot publish the manuscript, at least in its present form. However, if you can significantly strengthen the manuscript along the lines suggested by the reviewers, we will be glad to consider an adequately revised version of the manuscript for publication in *eLife*.

Essential revisions:

1) The first part of the study presents some high quality experimental work with the tools available in starfish, and the second part of the study presents an interesting model. However, the main weakness of this manuscript is that the model is not really constrained by the experimental work, and that the prediction of the model was not challenged by some additional experimental work (since the putative cross-linker was not identified). Thus, the study would be much stronger if the authors could experimentally test their model. At minimum, alternative models to explain the data, and the limitations of the study, should be extensively discussed.

2) A central argument for favoring a disassembly driven mechanism over a motor driven mechanism is the comparison of the fits in Figure 4 and Figure 6 to a viscoelastic model assuming either motor-driving or disassembly. Since this is so essential, the authors should give an expression for the contraction rate α(t) for the two models in the main text and make clear which measured quantities go into the fit and what other parameters are fitted. Importantly they should also give the value of all the fitting parameters. The authors should compare these parameters wherever possible to measured or measurable values (e.g. for filament length and depolymerization rate). In the current version, it is unclear if and how much these parameters vary from fit to fit and if they are within reasonable limits.

3) Along this line: the Motor Model depends on the actin concentration. This concentration does not change a lot for various UtrCH concentration (Figure 6). Yet, the fit of the 'low' to 'high' UtrCH concentrations (dashed red, green, blue lines, Figure 6) is drastically different from the control (black dashed line). Where does this sudden change come from if the input is so similar?

4) It appears that the only difference between the two viscoelastic models (D versus M) is the σ_c term. For the Motor-driven model there is a linear or quadratic dependency of A (actin density), for the disassembly driven model there is a linear dependency of A. Furthermore σ_c^D depends on k_depol and <l>. At first sight it appears surprising how this could lead to the large differences in the fits in Figure 6, but A (and perhaps also <l>?) are time dependent. It would be helpful to explain in more detail which variables are time dependent, and also to show <l> (t) in real units (i.e. micrometers). Is k_depol also a function of time?

5) Comparing the two models: Model M and D both get the actin concentration A as input (linearly or quadratically). However, in Figure 6 the two fits go into different directions. While in M the fit increases faster and gets a peak, the fit for model D becomes smaller in value and flatter. This suggest that the bigger effect on the fit of model D is given by the <l> and k_depol parameters. Again, it is important to give the value and potentially even plot these two parameters for the different cases. This would be also informative in the light of the claim that these quantities are gradually changed with the UtrCH injection.

[Editors' note: further revisions were requested prior to acceptance, as described below.]

Thank you for resubmitting your work entitled "A disassembly-driven mechanism explains F-actin-mediated chromosome transport in starfish oocytes" for further consideration at *eLife*. Your revised article has been favorably evaluated by Andrea Musacchio (Senior editor) and three reviewers, one of whom is a member of our Board of Reviewing Editors.

The manuscript has been improved but there are some remaining issues that need to be addressed before acceptance, as outlined below:

This is a revised version of a manuscript, which provides evidence that actin filament disassembly drives the contraction of a three-dimensional actin network that collects chromosomes and transports them to the forming microtubule spindle in starfish oocytes. The authors have satisfactorily addressed the majority of the previous concerns, and the manuscript is significantly improved. Also, the experimental and simulation details are much better clarified now.

However, the formin inhibitor data were not considered particularly strong, and therefore we feel that you should tone down the conclusions concerning the role of formin(s) as a cross-linker in this process (please, see specific comments below). All proposed changes can be addressed textually and do not require new experiments.

Specific comments:

1) The authors used SMIFH2 for inhibiting formins. To justify the use of SMIFH2 inhibitor, they referred to (Ucar et al., 2013). However, characterization of the effects of the inhibitor in this study were not particularly thorough, and this should be considered in the Discussion.

2) Even if the inhibitor works against formins, the effect of the inhibitor on the observed actin dynamics could be indirect e.g. due to altering the balance between Arp2/3 complex and formin -driven actin assembly pathways (see e.g. Burke et al., 2014). This possibility should be considered in the interpretation of the results.

3) In subsection “*In silico* reconstitution of the contractile system predicts a ‘depolymerization harnessing factor’ essential to drive contraction”, the authors test if the combination of disassembly and crosslinking could drive network contraction. This has already been done on a dynamic actin network (see: Ennomani et al., 2016), and should be discussed in the manuscript.

4) In the same subsection, the authors mention that mDia1 is able to track depolymerizing filament ends. However, these in vitro experiments (Jegou et al., 2013) were performed under specific conditions, where the critical concentration of actin was below the barbed end Cc. Thus, this argument cannot be used here.

---

## [Author Response]

Essential revisions:1) The first part of the study presents some high quality experimental work with the tools available in starfish, and the second part of the study presents an interesting model. However, the main weakness of this manuscript is that the model is not really constrained by the experimental work, and that the prediction of the model was not challenged by some additional experimental work (since the putative cross-linker was not identified). Thus, the study would be much stronger if the authors could experimentally test their model. At minimum, alternative models to explain the data, and the limitations of the study, should be extensively discussed.

In addressing this point, we made two major changes to the manuscript that we believe have significantly improved the revised version:

First, we explain more clearly how the parameters of the theory are constrained by experimental data (see also our response to the point below). We also changed the phrasing and expanded the text at several places to explain how developing the theory during this research has motivated experiments, and thus the model had a guiding virtue. Specifically, it motivated experiments in which we attempted to gradually increase and decrease disassembly rates.

Second, we now include new experiments aimed at identifying the ‘disassembly harnessing factor’ predicted by our simulations. We previously discussed that formin proteins in principle could be this factor as they do have the required activities. In the revised manuscript, we tested this experimentally by using a generic inhibitor of all proteins containing the conserved FH2 formin domain (SMIFH2). Intriguingly, this caused a significant reduction of disassembly rates and concomitant slowing of contraction, consistent with the predictions of the theory. In our view, these new findings constitute a significant advance and open a very exciting new perspective for a yet unexplored mechanism of contraction. However, the tools we have available in the starfish oocyte system are limited, and therefore identification of the specific formin-like protein is unfortunately not feasible in the frame of the present study. These limitations are now discussed explicitly in the manuscript.

2) A central argument for favoring a disassembly driven mechanism over a motor driven mechanism is the comparison of the fits in Figure 4 and Figure 6 to a viscoelastic model assuming either motor-driving or disassembly. Since this is so essential, the authors should give an expression for the contraction rate α(t) for the two models in the main text and make clear which measured quantities go into the fit and what other parameters are fitted. Importantly they should also give the value of all the fitting parameters. The authors should compare these parameters wherever possible to measured or measurable values (e.g. for filament length and depolymerization rate). In the current version, it is unclear if and how much these parameters vary from fit to fit and if they are within reasonable limits.

We expanded the revised manuscript and provide an improved supplementary document in which we clarify the model assumptions and how experimental data constrained both models. We took special care listing all input as well as fitting parameters used to derive the contraction rate αt over time, and we now show all equations in the main manuscript text.

Briefly, the viscoelastic model uses the following measured quantities:

a) We derived the *depolymerization rate, k* from the total fluorescence in the network to quantify relative changes in polymer mass over time (Figure 4; Equation 5; see Table 2).

b) We assume that filaments depolymerize with a constant speed from onset of contraction to the point at which the network would entirely vanish (Figure 1). Assuming that the distribution of filament length follows a Gaussian distribution, we can calculate from this relation the *mean length* as a function of time, and also the *number of filaments* as a function of time.

We then solve the partial differential equations from active-gel theory, with either a stress that is produced by motors, or a stress that depends on disassembly. For each model, we first obtained fitting parameters compatible with αt observed in control experiments (Figure 6—figure supplement 1, Table 1 and Supplementary file 2–Supplementary file 4). We then used the depolymerization rate *k* measured in perturbations and the fitting parameters from the corresponding control experiments to obtain the predicted contraction rate, αt. This αt is then compared to the experimentally measured contraction rate (Table 3 and Table 4). Unfortunately, all parameters of the model are renormalized, and we cannot evaluate experimentally the number of filaments, the length or the depolymerization speed. As a general evaluation of our theory, we show that similar results can be obtained by modeling the F-actin network as a purely viscous gel (Figure 6—figure supplement 1).

3) Along this line: the Motor Model depends on the actin concentration. This concentration does not change a lot for various UtrCH concentration (Figure 6). Yet, the fit of the 'low' to 'high' UtrCH concentrations (dashed red, green, blue lines, Figure 6) is drastically different from the control (black dashed line). Where does this sudden change come from if the input is so similar?

The major difference arises from the term of contractile stress σc in the two models: this is proportional to *A*^2^ in model M, and proportional to *A*^1^ in model D (Equations 8 and 9). Since UtrCH gradually stabilizes the F-actin network and thereby hinders actin disassembly, the F-actin concentration A is increasing with increasing UtrCH amounts. Consequently, the contractility increases much faster in model M, where the stress depends on *A*^2^, as compared to model D, where the stress depends on *A*^1^. This difference is qualitative, and is not rooted in the particular parameter values determined by the fit performed on the unperturbed/control experiments.

4) It appears that the only difference between the two viscoelastic models (D versus M) is the σ_c term. For the Motor-driven model there is a linear or quadratic dependency of A (actin density), for the disassembly driven model there is a linear dependency of A. Furthermore σ_c^D depends on k_depol and <l>. At first sight it appears surprising how this could lead to the large differences in the fits in Figure 6, but A (and perhaps also <l>?) are time dependent. It would be helpful to explain in more detail which variables are time dependent, and also to show <l> (t) in real units (i.e. micrometers). Is k_depol also a function of time?

We apologize for these omissions. In the revised manuscript, we define all variables and constants more strictly, and explicitly write for example *A*(r,t), *l*(r,t). The depolymerization rate *k* is an experimental measure derived from fitting the F-actin mass over time that does not depend on time except for Latrunculin A-treated oocytes. In response to acute Latrunculin A treatment, the depolymerization rate increases over time (Figure 7). As the reviewer points out, the main source of the differences seen on Figure 6 is the linear vs. quadratic dependence of stress on F-actin concentration.

5) Comparing the two models: Model M and D both get the actin concentration A as input (linearly or quadratically). However, in Figure 6 the two fits go into different directions. While in M the fit increases faster and gets a peak, the fit for model D becomes smaller in value and flatter. This suggest that the bigger effect on the fit of model D is given by the <l> and k_depol parameters. Again, it is important to give the value and potentially even plot these two parameters for the different cases. This would be also informative in the light of the claim that these quantities are gradually changed with the UtrCH injection.

The disassembly rate, *k*, the F-actin concentration, *A* and the normalized average length of F-actin *l* are given as inputs to the transport equations (Equations 11-13). The revised Supplementary file 1 offers a much-improved description of the theory. We describe the input parameters derived from experimental data as well as the fitting parameters used to derive the contraction rate αt over time. We provide tables to summarize the viscoelastic parameters used for both models M and D (Table 1 and Supplementary file 2–Supplementary file 4). We also provide values for the depolymerizarion rate *k*, derived from F-actin mass quantification in response to increasing amounts of injected UtrCH (Table 2, Table 4).

[Editors' note: further revisions were requested prior to acceptance, as described below.]

Specific comments:1) The authors used SMIFH2 for inhibiting formins. To justify the use of SMIFH2 inhibitor, they referred to (Ucar et al., 2013). However, characterization of the effects of the inhibitor in this study were not particularly thorough, and this should be considered in the Discussion.

We agree that the characterization of the inhibitor could have possibly been more thorough, and we have amended the Discussion accordingly with a few words of caution. However, the phenotype, failure of cytokinetic ring closure, observed by Ucar et al. (and confirmed by us) is very typical for formin inhibition across animal species, due to the conserved, essential role of the formin, mDia in cytokinesis. Ucar et al. observed the same phenotype following perturbations of mDia by other means, further supporting their conclusions.

Generally, the FH2 domain targeted by SMIFH2 is extremely well conserved across species, consistent with the original report showing specific effects from yeast to mammals (Rizvi et al., 2009), as well as in mouse oocytes by others (Kim et al., 2015).

2) Even if the inhibitor works against formins, the effect of the inhibitor on the observed actin dynamics could be indirect e.g. due to altering the balance between Arp2/3 complex and formin -driven actin assembly pathways (see e.g. Burke et al., 2014). This possibility should be considered in the interpretation of the results.

We fully agree with this comment, and we amended the Discussion accordingly, citing the above paper from the Kovar lab, as suggested. However, based on our previous data (Mori et al., 2014), we know that Arp2/3 is not primarily involved in nucleating filaments in the F-actin network transporting chromosomes.

3) In subsection “In silico reconstitution of the contractile system predicts a ‘depolymerization harnessing factor’ essential to drive contraction”, the authors test if the combination of disassembly and crosslinking could drive network contraction. This has already been done on a dynamic actin network (see: Ennomani et al., 2016), and should be discussed in the manuscript.

Ennomani et al., 2016 took a similar approach and used the Cytosim software to study the contractile response of a network in *absence* of filament dynamics. I.e. all their simulations were carried out with filaments of fixed length (see Supplemental Experimental Procedures, page 7 within the publication). In our opinion this is a critical difference, and correspondingly we felt no need to modify our discussion at this point.

4) In the same subsection, the authors mention that mDia1 is able to track depolymerizing filament ends. However, these in vitro experiments (Jegou et al., 2013) were performed under specific conditions, where the critical concentration of actin was below the barbed end Cc. Thus, this argument cannot be used here.

We fully agree with this comment, and we amended the Discussion accordingly. Unfortunately, we have no knowledge on the molecular composition, regulation and resulting properties of filament ends in the starfish oocyte. Therefore, it is difficult to evaluate whether and to what extent the findings under the specific in vitro conditions used by Jegou et al., 2013 are applicable to starfish oocytes.